Manuscript prepared for The Cryosphere
with version 2015/04/24 7.83 Copernicus papers of the LaTeX class copernicus.cls.
Date: 19 June 2017

# Geothermal flux and basal melt rate in the Dome C region inferred from radar reflectivity and heat modelling

Olivier Passalacqua[1,2], Catherine Ritz[1,2], Frédéric Parrenin[1,2], Stefano Urbini[3], and Massimo Frezzotti[4]

[1]Univ. Grenoble Alpes, LGGE, F-38401 Grenoble, France
[2]CNRS, LGGE, F-38401 Grenoble, France
[3]Istituto Nazionale di Geofisica e Vulcanologia, 00143 Roma, Italy
[4]ENEA, Centro Ricerche Casaccia, PO Box 2400, I-00100, Rome, Italy

*Correspondence to:* Olivier Passalacqua (olivier.passalacqua@univ-grenoble-alpes.fr)

**Abstract.** Basal melt rate is the most important physical quantity to be evaluated when looking for an old-ice drilling site, and it depends to a great extent on the geothermal flux (GF), which is poorly known under the East Antarctic ice sheet. Given that wet bedrock has higher reflectivity than dry bedrock, the wetness of the ice-bed interface can be assessed using radar echoes from the bedrock. But, since basal conditions depend on heat transfer forced by climate but lagged by the thick ice, the basal ice may currently be frozen whereas in the past it was generally melting. For that reason, the risk of bias between present and past conditions has to be evaluated. The objective of this study is to assess which locations in the Dome C area could have been protected from basal melting at any time in the past, which requires evaluating GF. We used an inverse approach to retrieve GF from the radar reflectivity at the bed. A 1D heat model is run over the last $800\,\mathrm{ka}$ to constrain the value of GF by assessing a critical ice thickness, i.e. the minimum ice thickness that would allow the present local distribution. A regional map of the geothermal flux was then inferred over a $80\,\mathrm{km}\times130\,\mathrm{km}$ area, with a N-S gradient, and with values ranging from $48$ to $60\,\mathrm{mWm}^{-2}$. The forward model was then emulated by a polynomial function, to compute a time-averaged value of the spatially variable basal melt rate over the region. Two main subregions appear to be free of basal melting because of the thin overlying ice, and a third one, north of Dome C, because of a low geothermal flux.

## 1 Introduction

### 1.1 The oldest ice research

Between 1.5 and 0.9 million years ago, the main periodicity of the global climate changed from $41\,\mathrm{ka}$ to about $100\,\mathrm{ka}$, as shown by temperature and sea level proxies retrieved in marine sediments (Mid-Pleistocene Transition, Lisiecki and Raymo, 2005). The causes of this major climate transition are still a matter of debate, and it would be easier to evaluate the different explanations proposed

by several authors (e.g. Raymo et al., 2006; Bintanja and Van de Wal, 2008; Martínez-Garcia et al., 2011; Imbrie et al., 2011) if greenhouse gases and other environmental proxies of this climatic period were analysed. Hence, retrieving an ice archive as old as 1.5 million years is one of the greatest challenges facing the ice core scientific community today (Brook et al., 2006).

The Dome C region (East Antarctica, see map in Fig. 1) has been of great interest to paleoclimate scientists in recent decades, and two deep ice cores have already been retrieved (Lorius et al., 1979; Jouzel et al., 2007), including the oldest one ever dated (800 ka, EDC ice core). The region has also been identified as possibly hosting even older ice (Fischer et al., 2013; Van Liefferinge and Pattyn, 2013). Unfortunately, these studies lacked precise information on the geothermal flux (GF) at the base of the ice. To ensure that the prediction of the age of a future old-ice core site is reliable, the thermal conditions at the base of the ice must be well constrained, because basal melting strongly affects the possible presence of old ice by continuously removing the oldest layers (Rybak and Huybrechts, 2008). In a dome region, the heat budget of ice depends mainly on the geothermal flux, which warms the body of ice from below, and on the temperature at the surface. But the resulting vertical temperature gradient also depends on the ice thickness, and, because the ice acts as an insulator, the thicker the ice cover, the warmer its base (Pattyn, 2010). Vertical cold advection due to movement of the ice also affects basal thermal conditions and must consequently be taken into account. Its value is linked to accumulation rate. Surface temperatures, ice thickness and accumulation rates have been directly measured, or their past values reconstructed, so the parameter about which the least is known regarding its influence on basal melting is geothermal flux.

In the Dome C region, significant subglacial bedrock features make the ice thin enough to for there to be a chance has been protected from melting. In the glaciology community, these locations are widely thought to be good old-ice candidates (Fig. 1 - A, B, C and D). In a context of locating the oldest-ice, our objective was to constrain the value of the geothermal flux and of the basal melt rate around Dome C. Later, this information will be used as a boundary condition for 3D mechanical simulations. As 3D models require many input parameters and are computationnally expensive, any additional information about the thermal regime of the ice that reduces parameters value range is welcome, and will help select a suitable drill site.

## 1.2 Assessing GF under ice sheets

The geothermal flux is usually derived from temperature gradients measured in boreholes in the ground, but this is not easy below ice sheets, where it is difficult to access to the bedrock. However, geothermal flux can be estimated from temperature profiles in ice core boreholes (Dahl-Jensen et al., 1998; Engelhardt, 2004). More precise results are expected for cold basal conditions, since uncertainties affect the basal melt rate for temperate basal ice (Grinsted and Dahl-Jensen, 2002), in which case only a minimum value of GF can be estimated.

As deep boreholes are not rare in East Antarctica, the value of the GF could be estimated from geological considerations (Pollard et al., 2005; Llubes et al., 2006), where uniform values were attributed to large geologically homogeneous areas. But spatial variability of GF occur at a much smaller scale, and efforts to model this variability have been made in the last decade. Two approaches based on geophysical information have been proposed to provide comprehensive maps at continental scale: one using a seismic model of the crust and upper mantle (Shapiro and Ritzwoller, 2004), and one a crustal thickness model derived from observations of the magnetic field (Fox Maule et al., 2005; Purucker, 2013). These studies show that the GF on the East Antarctic plateau is about $60 \pm 25\,\mathrm{mW\,m^{-2}}$, which unfortunately is much too coarse an estimation to give any precise value at a specific location. Moreover, Terre Adélie and George V Land could form part of the Mawson craton, which also forms the southern central part of Australia, where the GF shows high spatial variability (Carson et al., 2014). Even if the exact extent of this craton in Antarctica is not well known, in East Antarctica the GF could include hot spots, with a typical lengthscale of $10\,\mathrm{km}$.

Radio echo sounding (RES) measurements may help infer the basal conditions at regional scale, since the presence of water at the ice-bed interface pis responsible for a remarkable increase in the amplitude of the reflected echoes. For this reason, RES made it possible to detect melting at the base of ice sheets (Fujita et al., 2012; Oswald and Gogineni, 2012; Zirizzotti et al., 2012), as well as to map subglacial lakes (Siegert et al., 2005). But the presence of water is not sufficient to infer the GF, since water can originate either from local basal melting or be routed from elsewhere. Using a collection of water routing models, Schroeder et al. (2014) inferred the value of the GF needed to explain the observed pattern of radar echoes, and derived an average value with an uncertainty of $\pm 10\,\mathrm{mW\,m^{-2}}$ for the Thwaites Glacier catchment (West Antarctica).

Finally, basal melt rates for the region north of Dome C have been estimated by fitting the vertical strains with dated radar layers, to constrain the vertical advection of ice, and its energy budget (Carter et al., 2009). The uncertainty on the GF estimation was $\pm 12\,\mathrm{mW\,m^{-2}}$, which is a significant improvement over the previous estimations, but is still too large to identify a location with very old ice. What is more, their study area does not cover the main old-ice candidates located to the east and south-west of Dome C, so a new local estimation is needed. As dated layers are not available everywhere, we used a reflectivity-based approach like that of Schroeder et al. (2014), but adapted to the specific pattern of radar echoes under Dome C, where the melting point is not reached everywhere.

## 1.3 Exploitation of available RES dataset around Dome C

Based on radar equations, Zirizzotti et al. (2012) proposed a method to recognize wet areas using the Italian RES dataset collected around Dome C (http://labtel2.rm.ingv.it/antarctica/). These authors used a linear model of electromagnetic wave adsorption in the ice column, based on analysis of the EDC ice core, to account for the differences in amplitude (in dB) between echoes at the surface of the ice and at the bottom. The thresholds used to ascribe an echo to a wet or dry basal contact

were $\geq 7.7\,\mathrm{dB}$ and $< 1\,\mathrm{dB}$ respectively. Figure 1 shows the distribution of dry and wet areas under Dome C, which reveals interesting patterns: $(i)$ the Concordia trench is characterised by the presence of wet points only, which is a consequence of the very thick ice; $(ii)$ the tops of the two main bedrock reliefs (candidate $\mathtt{A}$ and candidates $\mathtt{B-C-D}$) appear to be dry; $(iii)$ almost all the northern points are dry, despite the very thick ice at this location; $(iv)$ in between, and in particular under the EPICA drilling site, dry and wet points are scattered, with no clear trend.

We would like to emphasise that these observations only refer to the present day bed conditions, whereas the historical conditions covering the glacial/interglacial periods need to be investigated to evaluate the quality of a future old-ice drilling site. The absence of basal water today means cold basal ice, but this may simply be a consequence of the very strong temperature signal of the Last Glacial Maximum (LGM) lagged by the thick ice. The present cold state may thus not be representative of all the glacial-interglacial periods, and thawing could have occurred in the past under different climatic conditions.

Hence, the aim of this work was to to constrain sites known to be frozen today and that are very likely to have also been frozen in the last $800\,\mathrm{ka}$, increasing the probability that very old ice has been preserved. The dataset available from Zirizzotti et al. (2012) provides informations on whether melting occurrs, but not on its amount and its temporal and spatial variations. The only way to obtain this information is first to retrieve the spatial distribution of GF and then to use it to infer basal melting over time.

Here we present a 1D heat model forced by reconstructed climatic conditions, which we ran in two ways. First, we solved an inverse problem with the model to infer the value of the GF in the Dome C region, using radar echoes as observational constraints. The pattern of wet and dry areas made it possible to estimate a critical ice thickness that corresponding to a threshold between frozen and thawing ice. For a given GF and vertical advection, this thickness is unique, so that in turn the distribution of GF can be inferred from the pattern of basal echoes. Second, we ran the model forward to compute the average past basal melt rate under the GF inferred at the first step. We present an easy-to-use emulator of the past basal melt rate, which is a convenient thermal boundary condition for future modelling works, and key-criterion in the location of old-ice drilling sites. For the sake of convenience, the domain is referenced using pairs of letter and numeral, corresponding to the grid of the Italian survey (Fig. 1). In particular, two promising old-ice candidates are located at C6 and H1.

## 2 Heat model

In this section we present the heat model accounting for the relationships between GF, ice thickness, vertical advection, and the temperature of the ice. The main assumptions under which the model is run are presented at the end of the section. The areas for which these assumptions are valid are presented in section 3.

### 2.1 Geometry and coordinate system

In a dome region, horizontal velocities are very small (a few centimetres per year, Vittuari et al., 2004), meaning horizontal advection terms can be neglected. Similarly, the deformational heat is several orders of magnitude smaller than the vertical advection term. Finally, due to the small aspect ratio (thickness/characteristic horizontal length), the temperature field is mainly vertically stratified, so horizontal diffusion can also be neglected. Thus, the heat balance is assumed to be only vertically dependent.

Here we consider a one-dimensional vertical domain, oriented upwards along the $z$-axis. Instead of being set in the $z$-coordinates, the equations are expressed in reduced depth $\zeta = (s - z)/(s - b)$, $s$ being the surface height, and $b$ the bed height. In the $\zeta$-coordinate system, the domain size remains the same whatever the ice thickness $H$, and no remeshing is needed in the resolution of the finite difference scheme. Therefore the ice thickness is a simple parameter evolving over time according to surface accumulation forcing. Changes in thickness are accounted for by the conceptual model of Parrenin et al. (2007), which, for Dome C, emulates the 3D large scale simulations of Ritz et al. (2001). At each timestep, for the thickness and the bedrock elevation, explicit expressions are solved, which depend on the accumulation rate and six tabulated parameters, which account for the sensitivity of physical quantities (in particular bedrock and surface height and ice thickness) to climate forcing. The typical difference in the ice thickness in the glacial and interglacial periods is $150\,\mathrm{m}$.

### 2.2 Heat equation

The heat balance of ice only depends on the vertical coordinate, and is written for the ice temperature $T$ as follows in the $\zeta$-coordinate system (Ritz et al., 1997):

$$\frac{\partial T}{\partial t} = \frac{1}{\rho c D H^2} \frac{\partial}{\partial \zeta} \left( K \frac{\partial T}{\partial \zeta} \right) - u_\zeta \frac{\partial T}{\partial \zeta} \tag{1}$$

where $K$ is the thermal conductivity of the material (firn or ice), $c$ its heat capacity, $\rho$ the density of ice ($917\,\mathrm{kg\,m^{-3}}$), and $D$ the relative density of the material ($< 1$ for the firn and 1 for the ice, dimensionless). The vertical velocity $u_\zeta$ accounts for the true ice velocity, but also for changes in the ice thickness, which in turn modifies the relationship between $z$ and $\zeta$.

#### 2.2.1 Ice thermal properties

The specific heat capacity and heat conductivity depend on the absolute temperature $T$ as follows (Cuffey and Paterson, 2010, p. 400):

$$c = 152.5 + 7.122 \cdot T \; [\mathrm{J\,K^{-1}\,kg^{-1}}] \tag{2}$$

$$K = 9.828 \cdot e^{-0.0057 \cdot T} \; [\mathrm{W\,m^{-1}\,K^{-1}}] \tag{3}$$

The presence of firn at the surface of the ice reduces conductivity in the upper part of the ice column, and this must be accounted for (Cuffey and Paterson, 2010, p. 401):

$$K = \frac{2K_i \times D}{3 - D} \tag{4}$$

where $K_i$ is the conductivity of the ice. The density profile of the Dome C firn layer is taken from
Parrenin et al. (2007).

### 2.2.2   Basal boundary conditions

At the ice/bed interface, the thermal conditions depend on whether or not the pressure melting point is reached. For thawing glacier ice, the melting temperature $T_m$ linearly depends on the ice pressure $P$ and the partial pressure of air dissolved in the ice $P'$, which is expressed as (Ritz, 1992):

$$T_m = 273.16 - 0.074 \cdot P - 0.024 \cdot P' \tag{5}$$

In ice sheets, $P'$ is of the order of $1\,\mathrm{MPa}$. This expression is compatible with the temperature profile at the EDC borehole, where the deepest measured temperature was $270.05\,\mathrm{K}$ at $3223\,\mathrm{m}$, $50\,\mathrm{m}$ above the bedrock. The temperature profile can be extrapolated to the bedrock (in the same way as Dahl-Jensen et al. (2003) at North GRIP) to $271.04\,\mathrm{K}$, where Eq. (5) gives $270.96\,\mathrm{K}$. As
the ice in the dome region moves very slowly, we assume that the pressure only depends on ice thickness. Once the temperature field is computed, the melt rate $m$ at the bottom can be explicitly known by

$$m = \frac{1}{\rho L} \left( \Phi_g - \frac{K}{H} \left. \frac{\partial T}{\partial \zeta} \right|_{\zeta=1} \right) \tag{6}$$

where $\Phi_g$ is the GF and $L$ the latent heat of ice ($\mathrm{J\,kg^{-1}}$). In this equation, no friction heating is
considered, since the basal velocities are very small, and the corresponding additional heat source is a second-order term (Krabbendam, 2016). For cold basal ice, Eq. (6) is used as a Neumann boundary condition, equating $m$ to zero.

### 2.2.3   Boundary condition at the ice surface

The atmospheric temperature forcing is continuously transferred through the whole ice column, so
the present thermal conditions at the bed are the result of the entire climate history. The atmospheric

paleotemperatures have been estimated over the last $800\,\mathrm{ka}$ from the $\delta\mathrm{D}$ measurements made on the EDC ice core (Jouzel et al., 2007). Here we use the model and notations of Parrenin et al. (2007), linking the deviation of the deuterium content $\Delta\delta D$ of the ice to the surface temperature $T_s$ w.r.t. a reference temperature $T_s^0$:

$$T_s = T_s^0 + \alpha\Delta\delta D \qquad (7)$$

where $\alpha$ reflects the amplitude of past changes in $T_s$. The influence of the value chosen for $\alpha$ has to be examined, since uncertainties affect our knowledge of the isotopic thermometer at long timescales in Antarctica, and $\alpha$ can vary from -10% to +20% (Jouzel et al., 2003). In addition to the nominal value of $\frac{1}{6.04}$ K, we will perform sensitivity studies with two additional values of $0.13$ and $0.20$, chosen

so that the maximum temperature difference over climatic periods with the nominal run is $\pm2\,\mathrm{K}$. The corresponding accumulation rates are also taken from Parrenin et al. (2007), who considered an exponential accumulation model, linking the accumulation rate $a$ to a reference accumulation rate $a^0$:

$$a = a^0\exp(\beta\Delta\delta D) \qquad (8)$$

where $\beta$ reflects the amplitude of past changes in $a$. For Dome C, the value of the $\beta$ coefficient was evaluated at $0.0156\pm0.0012$ using an inverse method constrained by known age markers. We consequently checked the sensitivity of our model for three values of $\beta$ ($0.0138$, $0.0156$ and $0.0175$). The two extreme values were chosen so that the maximum accumulation difference over climatic periods with the nominal run is $\pm0.25\,\mathrm{cm\,a^{-1}}$, which is a more intuitive way to express the sensitivity.

Regarding the choice of the initial state, we considered that the duration needed for a step climatic signal to reach the bedrock is $\sim 10\,\mathrm{ka}$, plus $\sim 100\,\mathrm{ka}$ to stabilise. As we do not know the true initial state of the temperature profile in the past, we decided to run the model over the whole reconstructed period ($800\,\mathrm{ka}$), so that the computation is independent of the initial state and the final vertical temperature profile is as realistic as possible.

## 2.3  Velocity model

In the heat balance, the vertical advection of ice acts by transporting cold towards the bed. Instead of solving the equations of motion, it will be accounted for by a 1D shape function $\omega$ (Parrenin et al., 2007; Ritz, 1987):

$$\omega(\zeta) = 1 - \frac{p+2}{p+1}\cdot\zeta + \frac{1}{p+1}\cdot\zeta^{p+2} \qquad (9)$$


$$u_z = -(a - \frac{\partial H}{\partial t} - m)\cdot\omega(\zeta) - m \qquad (10)$$

where $u_z$ is the vertical velocity of ice in the $z$-coordinate system, $a$ is the surface accumulation rate and $m$ is the basal melt rate, which is considered to be positive when melting. The shape parameter $p$ influences the temperature profile, since it controls the vertical advection of ice from the surface to the base. Far from divides, and in the case of isotropic ice, this parameter depends on the non-linearity of the ice rheology and on the vertical temperature gradient at the base (Lliboutry, 1979):

$$p = n - 1 + \frac{Q}{RT_b^2} \left. \frac{\partial T}{\partial \zeta} \right|_{\zeta=1} \tag{11}$$

where $n$ is the exponent of the Glen's flow law, $Q = 60\,\text{kJ}\,\text{mol}^{-1}$ is an activation energy, $R = 8.314\,\text{J}\,\text{mol}^{-1}\,\text{K}^{-1}$ the gas constant, and $T_b$ is the basal temperature. Following Eq. (11), the values of $p$ should theoretically range between 7 and 9 on the East Antarctic plateau. But in practice we will use $p$ close to divides in a wider range of values, as a parameter able to account for realistic vertical velocity profiles. For exemple, dome profiles are expected to correspond to low $p$ values due to Raymond arches (Raymond, 1983), whereas basal sliding makes the profile more linear and increase the value of $p$. Above a typical value of $p = 10$, the profile is very close to being linear (Fig. 3), and higher values of $p$ do not significantly change the profile. In the case of a shape function in a dome region (Parrenin et al., 2007), the vertical velocity $u_\zeta$ in the $\zeta$-coordinate system is then expressed as follows:

$$u_\zeta = \frac{1}{H} \left( \frac{\partial H}{\partial t}(1 - \zeta - \omega(\zeta)) + m(1 - \omega(\zeta)) + a\omega(\zeta) \right) \tag{12}$$

which is equivalent to

$$u_\zeta = \frac{1}{H} \left( -u_z + (1 - \zeta)\frac{\partial H}{\partial t} \right). \tag{13}$$

## 2.4 Proceeding assumptions

In addition to the 1D geometric assumption allowed by the proximity of the dome, two further assumptions are made in this paper, and are now justified. First, as no routing model is used to make the basal water circulate at the ice-bed interface, this implicitly means that the presence of water at the bed corresponds to a thawing basal ice. Second, GF is assumed to be spatially uniform on short spatial scales.

### 2.4.1 Correspondence between wet and thawing areas

Since the wetness indicated by the basal echoes will be used as constraints in the inverse model, we first have to identify the origin of the observed basal water. Either the local ice is temperate and the water comes from locally thawed ice, or the local (cold) ice energy budget was upset by the

latent heat of water flowing from a temperate location, along the gradient of hydraulic potential. In the Dome C area, the surface of the ice is very flat and the hydraulic potential almost follows the bedrock heights, which correspond to the local maxima of hydraulic potential. These topographic features prevent water from moving upwards (Fig. 2), so the water observed on the lee of a relief, close to the transition with dry areas, must be of local origin.

Could the dry areas in Fig. 1 correspond to thawing ice whose meltwater has flowed away? Meltwater can be driven out by different types of hydrological networks: connected cavities (Lliboutry, 1968; Kamb, 1987) or an efficient network of structured channels (Röthlisberger, 1972). Weertman (1972) showed that these channels cannot form upstream of the hydrological network, which is the case here, and would rather take the form of a film of water. Whatever the exact type of structure (film or cavities), basal melting is a permanent process that feeds the hydrological network. A continuous hydrological network upstream is only fed by local melting, so that, unless the network is disconnected, the thawed water cannot be driven out faster than the melt rate. As a consequence, some water always remains at the base of the melting ice. Furthermore, water at the base enables basal sliding, and reduces the basal drag. For a given ice flux, the surface slope gives some indication on the relative importance of internal deformation and basal sliding in the ice motion. Local steeper surface may be associated with more basal drag and ice deformation, whereas local flatter surface may be associated with more sliding. Most of the spots where the model will be run are located on slopes which are locally steeper than the regional slope (Table 1), meaning it is likely that sliding does not occur there, an additional clue that the base is cold. As a consequence, we suggest that the dry areas in Fig. 2 probably do correspond to cold ice.

### 2.4.2 GF spatial variability

Based on the little that we know about the geology, we can assume that the GF is uniform on a $\sim 10\,\text{km}$-lengthscale. Under this assumption, the presence of basal water is only the consequence of thicker ice, and not of spatial variations in GF. As such, thickness and reflectivity have to be locally correlated, and the 1D model will be run for small areas where this correlation is reasonable (section 3).

### 2.5 Numerical method

Equations were solved with a finite difference implicit scheme on a 50-element regular mesh, and the temporal dependence of the model was solved at a 1000-year timestep. These values were selected as trade-offs between obtaining accurate results and computation speed. The dependence of the basal temperature on discretization is limited to a few tenths of $\text{K}$.

The physical coupling of flow and heat content of the ice is accounted for by non linear iterations, in which updated values of $m$, $K$ and $c$ are computed, until the temperatures no longer show a discrepancy larger than $10^{-5}\,\text{K}$ between two iterations. At each timestep, the heat equation is solved

for a boundary condition corresponding to the thermal state of the basal ice (temperate or cold). The computed solution may be inconsistent with the imposed boundary condition, i.e. that the pressure melting point may be exceeded (cold $\rightarrow$ temperate), or the melt rate may become negative (temperate $\rightarrow$ cold). In these cases, the equation is solved again with the new consistent boundary condition.

### 3 Spots where GF will be inferred

We defined the critical ice thickness $H_c$ as the minimum ice thickness at which present basal melting becomes possible for a given GF $\Phi_g$ and a given shape parameter $p$. To determine $H_c$, the model was run with increasing values of the GF, until the melting point was reached at present, so that we we were able to determine a unique tuple $(H_c, p, \Phi_g)$. As the unknown is the GF, an estimation of this critical ice thickness was needed, and this was done in two steps from the wet/dry map in Fig. 1.

First, we identified spots at which ice thickness and reflectivity are correlated, i.e. for which the top of bed reliefs are dry and their lees are wet (see Fig. 2). Melting starts to become physically possible between the two. Ten corresponding spots were selected (black rectangles in Fig. 1), where reflectivity and ice thickness are somehow correlated, and that are hereafter denoted by the indexes of their central point (Fig. 4). For two spots (E4 and E6), the correlation was weak, but we kept them as they are the only ones available in the central part of our study area.

Second, we evaluated the critical ice thickness for each spot. Along the radar line, moving upwards along a bed relief, we selected the wet point at which the ice is the thinnest, and the dry point at which the ice is the thickest. The same is done moving downwards, so that four points were selected along the radar line (black arrows on Fig. 2). The critical ice thickness $H_c$ was estimated by the average of the ice thickness measured at these points. The uncertainty on $H_c$ was taken as half the standard deviation of the four thicknesses, so that almost all the possible values ($H_c \pm 2\sigma$) ranged within the extreme thicknesses measured at the spot considered. For the large spots C3 and C6, this operation was performed twice, on the two perpendicular radar lines going through them (radar lines C and 3, and C and 6 respectively), so that eight dry/wet points were selected.

### 4 GF inversion

The two main input parameters that influence the presence or absence of water on the bed are the ice thickness $H$ and the shape parameter $p$, both of which are affected by uncertainties. The heat model was run for 200 $(H, p)$ values spread along Gaussian distributions, so that a posterior distribution of $\Phi_g$ was put out (Tarantola, 2005).

The critical thickness $H_c$ measured on the map was used directly as a prior for $H$. The shape parameter $p$ cannot be less than $-1$, so the inversion was done on the derived parameter $p'$:

$$p' = \ln(p+1) \tag{14}$$

The prior value for $p'$ was taken as a Gaussian distribution of mean $\bar{p}' = 1.5$ and standard deviation $\sigma_{p'} = 0.3$ so that the corresponding values for $p$ were mainly distributed between common values of 1 and 10. For a given present-day critical ice thickness $H_c$ and a given $p$, the 1D heat model was run for increasing values of $\Phi_g$ (by steps of $0.25\,\mathrm{mW\,m^{-2}}$), and the first $\Phi_g$ value that allowed melting at the present time was selected as the local GF. The thicker the ice, the lower the GF needed to allow for melting. All the GF values were in the range between 40 and $70\,\mathrm{mW\,m^{-2}}$.

## 5   Results

### 5.1   Inverse mode: geothermal heat flux

The inferred values of the mean GF for the ten measurement spots ranged between 48 and 60 $\mathrm{mW\,m^{-2}}$ (Table 1), and the highest values of the GF were found south of Dome C. Given the model assumptions, the inferred value for the GF at two potential drilling sites (C6 and H1) were respectively $59.3 \pm 2.2$ and $53.9 \pm 3.3\,\mathrm{mW\,m^{-2}}$.

A kriging interpolation was used to evaluate the GF between the spots evaluated. Unfortunately, the spots are scarce (there are only ten points), and are unevenly distributed throughout the study area. As such, the computed experimental variogram was poorly described, and we had little confidence in the computed kriging standard deviation. Moreover, the possibility of local variability at a scale of a few tens of kilometres cannot be excluded, so that the validity of such an interpolation is limited. Nevertheless, we underline the importance of such a map to show the regional trend that can be expected around Dome C.

Figure 5 shows the interpolation between the ten spots, in which the GF field evolves smoothly along a N-S gradient, with a typical norm of $0.1\,\mathrm{mW\,m^{-2}\,km^{-1}}$. At the two sites where the correlation between reflectivity and ice thickness was weak (E4 and E6), the inferred values of the GF nevertheless matched the N-S gradient, even if the value at E4 appeared to be significantly lower than its neighbours. Given that the kriging standard deviation does not account for the uncertainty on the GF estimation at the spots, the GF at the EPICA drilling site was estimated at $54.5 \pm 3.5\,\mathrm{mW\,m^{-2}}$.

### 5.2   Forward mode: emulator of the basal melt rate averaged over the last 400 000 years

A time-averaged value of the basal melt rate has to be computed to assess the risk of the oldest ice layers being lost during the glacial/interglacial periods. To do so, the GF field inferred at the previous step was used as an input for the heat model. Running it transient enabled the computation of the

basal melt rate at each time step, which was done for given values of $H$, $\Phi_g$ and $p$, whereas $\alpha$ and $\beta$ were kept at $1/6.04$ and $0.0156$ respectively. Because of the uncertainty on the initial state, it was not

clear if our temperature profile for the first glacial cycles was accurate, so we averaged the computed melt rate over the last $400\,\mathrm{ka}$ only.

As computing a given set of parameter takes several minutes, computing the basal melt rate for each point of the Dome region would be far too expensive. Here, the result of the whole forward model is mimicked by an emulator that depends on the input parameters $H$, $\Phi_g$ and $p$. The empir-

ical relationship linking the average melt rate to the model parameters appears to be very regular (Fig.6). The slope of the isomelt contour lines shows equivalence between GF and ice thickness ($1\,\mathrm{mW\,m^{-2}}$ roughly corresponds to $60\,\mathrm{m}$). The sensitivity of the melt rate on $p$ ($1 \le p \le 10$) appears to be equivalent to a range of $2\,\mathrm{mW\,m^{-2}}$ of GF. An additional flux of $1\,\mathrm{mW\,m^{-2}}$ correspond to an increased melt rate of $0.09\,\mathrm{mm\,a^{-1}}$. To account for this relationship by a multivariable poly-

nomial, it seems natural to make $m$ depend linearly on $\Phi_g$, and quadratically on $H$ and $p$. Over the positive-melt-value domain, we used a least-square minimization method to compute the following relation, which is valid for the ranges of values in figure 6:

$$
\begin{aligned}
m = -\,& 5.148 \times 10^{-7} H^2 + 4.688 \times 10^{-3} H + 89.519 \Phi_g \\
& + 3.08 \times 10^{-3} p^2 - 5.887 \times 10^{-2} p - 14.335.
\end{aligned}
\tag{15}
$$

The performance of the emulator was assessed by comparing the melt rate output from the thermal model with the one computed with the emulator. The average absolute error was $0.014\,\mathrm{mm\,a^{-1}}$, and the corresponding standard deviation was $0.016\,\mathrm{mm\,a^{-1}}$, so that the errors due to the emulator were significantly lower than the corresponding uncertainties due to the GF and $p$. This polynomial function was thus considered to be sufficiently precise for our purpose. Here we emulated the values

of $m$ with $H$ given by the Bedmap 2 dataset, but when a refined bedrock dataset is available, this emulator will be easy to use to compute updated estimations of the basal melt rate.

Figure 7 shows the basal melt rate corresponding to the GF field interpolated from the central values of $\Phi_g$ inferred at each measurement point (called $\hat{\Phi}_g^m$), and the central value of $p'$ ($p = 3.5$). The basal melt rate increased to more than $0.6\,\mathrm{mm\,a^{-1}}$ where ice is very thick, and vanished over

several spots. The candidates A, B, C and D appeared to be melt-free, which was expected, since they benefit from a relatively thinner ice. Less expected was the presence of a potentially melt-free area sixty kilometres north from Dome C (point N8 in Fig.1). The ice is quite thick there ($\sim$3300 m), but the computed GF is low enough to prevent ice from melting. Changes in the basal melt rate over time are also presented for two ($H_c$, $\Phi_g$) temperate configurations, leading to the same average

value (Fig. 8). The difference between the minimum and maximum value of the basal melt rate was $0.6\,\mathrm{mm\,a^{-1}}$ for 2770 m-thick ice, and an additional 500 m of ice dampens this amplitude by half, and smooths the high frequencies. The difference between present melt rates and maximum past values

suggests that, more generally, it is possible that present cold basal ice melted during the warmest periods.

## 5.3 Sensitivity tests

Next, we investigated the influence of climatic forcing, by running the model with different slopes of the isotopic thermometer. The results for the GF obtained using the inverse method shifted positively by $1.4 \, \text{mW} \, \text{m}^{-2}$ for $\alpha = 0.20$ and negatively for $\alpha = 0.13$. However, the average basal melt rate is changed by $0.1 \, \text{mm} \, \text{a}^{-1}$ in the reverse direction, so that the final difference in the inferred melt rate was only $0.04 \, \text{mm} \, \text{a}^{-1}$ (Table 2). Because parameter $\alpha$ affects both the derivation of the GF at the spots and the forward model melt rate, the two steps partly offset one another when producing the melt rate map.

For extreme values of the $\beta$ coefficient (accumulation model), the GF results shifted positively by $0.3 \, \text{mW} \, \text{m}^{-2}$ for $\beta = 0.0138$ and negatively for $\beta = 0.175$. Like for $\alpha$, the average basal melt rate shifted in the reverse direction by $0.01 \, \text{mm} \, \text{a}^{-1}$, so that the final difference in the inferred melt rate was only $0.02 \, \text{mm} \, \text{a}^{-1}$.

The accumulations reconstructed at the dome itself were spread equally across the whole region, whereas in pratice spatial variations surely affect the surface accumulation rate around Dome C (Frezzotti et al., 2005). The sensitivity of our model to a $\pm 10 \, \%$ variation of the surface accumulation resulted in a $\pm 0.3 \, \text{mW} \, \text{m}^{-2}$ difference for the GF, and to an opposite $\pm 0.08 \, \text{mm} \, \text{a}^{-1}$ for basal melt rate. The final sensitivity of the basal melt rate was thus $\pm 0.05 \, \text{mm} \, \text{a}^{-1}$. A higher accumulation rate results in a lower basal melt rate, which is expected.

Given that the expression of the pressure melting point is an unusual choice in glaciology (Eq. 5), the inverse method was reiterated with a more common value corresponding to saturated air in the ice ($T_m = 273.16 - 0.098 P$, Cuffey and Paterson, 2010) as a test. The inferred GF values were found shift by $-0.6 \, \text{mW} \, \text{m}^{-2}$ compared to our expression. The results in terms of basal melting were not significantly affected by the expression of the pressure melting point, nor was the regional pattern of GF.

## 5.4 Forward mode: emulator of the critical ice thickness

One way to assess the performance of our model is to compare observed basal state (wet or dry, Zirizzotti et al. (2012)) with the model simulation for present time. For $p = 2$, we computed the $(\Phi_g, H_c)$ relation, corresponding to the present basal state, by sampling $H_c$ between $2\,700 \, \text{m}$ and $3\,300 \, \text{m}$. The two parameters are linked by the following empirical relation:

$$H_c = 1013272.4 \Phi_g^2 - 170906 \Phi_g + 9486.0 \tag{16}$$

This made it possible to compute the difference between critical ice thickness and the present ice thickness (Bedmap2), so that negative values correspond to melting areas, and positive values to melt-free areas. Our aim was to broadly mimic the wet/dry pattern in areas where no critical thickness had been measured (north of Dome C). To facilitate the comparison, we slightly tuned the imposed GF field, within the uncertainty range. The map in Fig. 9 was built with $\hat{\Phi}_g^m - 1\,\mathrm{mWm}^{-2}$.

Superimposed on the observed data, the model output shows that large-scale patterns of wet-dry areas were respected, especially on steep bed slopes (candidate B,C,D, and to a lesser extent candidate A). However, at certain points on these bed reliefs, there was a discrepancy between model outputs and observed data, but the gap in the critical thickness was often close to $0\,\mathrm{m}$ (D3, D5, D8, M3), meaning that a small change in GF forcing, or a better description of the ice thickness, would assign these particular points more accurately. The $1\,\mathrm{km}$-resolution of the Bedmap 2 bedrock dataset (Fretwell and coauthors, 2013) smoothed the along-track subkilometric features detected by our RES survey.

The steeper the bed, the sharper the limit between melting and non-melting areas. In the central, flatter, part of the study area, the position of this limit was more blurred. As we were unable to assess the GF in that par of the study area using our method, it was interpolated. Despite the lack of constraints, several small-scale features are well mimicked (dry areas at I9, G-H8, wet areas at G9 and L7). Other regions were not attributed in agreement with observations (G6-7-8, H-I8), meaning that the GF is overestimated, probably up to $3\mathrm{mW\,m}^{-2}$, which is consistent with the uncertainties produced by our method (inversion and interpolation).

The heat model allows easy detection of the very strong climatic signal of the LGM when it reaches the bed, and the corresponding time lag $\Delta t$ for basal temperature appears to be linearly dependent on the ice thickness in the range 2700-3500 m by the following relation

$$\Delta t = 10H - 16610. \tag{17}$$

The isocontours of time lag $\Delta t$ are simply parallel to the thickness contours, but enable a temporal interpretation of the wet/dry patterns (Fig. 9). Around $10\,\mathrm{ka}$ are required for the signal to reach the tops of bed reliefs, but almost twice that for $3\,400\text{-}\,\mathrm{m}$-thick ice. Hence, considering the duration of deglaciation, depending on the ice thickness the thermal state of the basal ice may correspond to very different climatic periods.

## 6 Discussion

### 6.1 Consistency with published data and measurements

Our GF values are comparable to those inferred by Fox Maule et al. (2005) and Purucker (2013), but, because the uncertainties of their method are quite large, they cannot be considered as a reliable

comparison. For the northern part of our study area, the relatively low values of GF ($\sim 50\,\mathrm{mW\,m^{-2}}$ or less) are consistent with those estimated by Carter et al. (2009) at the same place, except on a bed relief (candidate D), where they found a positive heat flux anomaly and a high melt rate on its flanks ($\sim 2.5\,\mathrm{mm\,a^{-1}}$). The heat anomaly was not induced with our method, since the GF is interpolated at this location. This shows that our method is reliable at locations where we do have an estimation of the critical thickness (corresponding to potentially interesting dry, low-reflective cold spots), and elsewhere at least describes a GF regional background. The GF field is also compatible with the values inferred by (Siegert and Dowdeswell, 1996) to model the temperature above subglacial lakes around Dome C ($41 - 58\,\mathrm{mW\,m^{-2}}$). But their values were only minima compatible with ice at the pressure melting point, whereas the present work benefits from the dry areas detected by RES used as upper bounds constrains.

At the EPICA drilling site, we also computed an average melt rate of $0.32 \pm 0.25\,\mathrm{mm\,a^{-1}}$, and this value seems to be slightly lower than, but yet consistent with the value of $0.56 \pm 0.19\,\mathrm{mm\,a^{-1}}$ previously inferred by Parrenin et al. (2007), with the inverse model used for dating the ice core. The range of possible basal melt rates inferred with our method seems sufficiently realistic to contain its effective local value, at a location where no observed critical thickness is yet available. The basal melt rate inferred for the different old-ice candidates, for which we had observations of $H_c$, is thus reliable.

## 6.2 Model assessment

### 6.2.1 Method validity

Around Dome C, ice may flow over a hilly bedrock, and the velocity profile is not necessarily as smooth as a 1D synthetic shape function. However, the energy budget at the base of the ice actually depends on the total advection of cold from the top, not on small scale variations in the vertical velocity profile. Using a synthetic profile is just a convenient way to account for a realistic vertical advection of ice towards the bed, controlled by a single parameter, $p$.

Some of our confidence intervals are quite low (E4, E6, L7 and N8), one consequence of the tiny difference in altitude measured on Fig. 1 between the highest wet points and the lowest dry points at a given spot. Since the correlation between ice thickness and reflectivity was weak, the confidence intervals at E4 and E6 are probably underestimated, and some local effects (e.g. small relief and GF variabilities) may not have been accounted for in this study.

We cannot exclude assignment errors in the dry points, if a small water film is present but was not detected. If so, we would only be able to assess a lower boundary for the GF. However, the absence of detection of the meltwater likely means that the water film is very thin and the melt rate is very low. The average inferred GF would only be offset by a small amount, and the regional GF gradient would remain unchanged.

### 6.2.2 Sensitivity to parameters

The sensitivity of our results to climatic forcing shows that the confidence interval on the GF at the 10 spots could be larger than presented so far, possibly by $\sim 2\,\mathrm{mW\,m^{-2}}$. However, for the archiving process, the truly important parameter is the basal melt rate, which is much less sensitive. For example, a lower $\alpha$ corresponds to a lower inferred GF (inverse run), but the average melt rate is higher for the same GF (direct run). The latter compensates for two thirds of the effect of the former, so that the melt rates computed for the ten measurement spots are quite robust to our lack of knowledge on climate forcing.

As the present surface accumulation pattern shows a N-S gradient, we wonder if our GF gradient could be due to an artefact of our method, which does not account for the spatial variations in accumulation. The sensitivity of the GHF on the surface accumulation is less than a few tenths of $\mathrm{mW\,m^{-2}}$ so that, accounting for its spatial variations would not radically modify our results. Furthermore, we do not know if the pattern of accumulation over the glacial/interglacial periods remained stable, so assuming its stability would result in unnecessary additional uncertainties.

### 6.2.3 Spatial variation in the GF field

In our point-by-point method, there are no horizontal regularization constraints to make the GF realistic at a regional scale. Yet, the inferred heat flow at the ten spots reflects a certain spatial pattern, which is a good sign of plausibility. More fundamentally, at first order, we can explain a relatively complex pattern of wet/dry areas in a whole region with a single physical key (a smoothly-varying GF field) with no 2D water routing model, or horizontal heat transport depending on both ice flow and bedrock topography. This means that the main physical mechanisms have been taken into account, at least where it was possible to evaluate the critical thickness, on significant topographic features. Where the GF is interpolated, in flatter areas, the lack of constrains prevents us from really assessing the validity of the method, meaning that our method needs a sufficient hilly bedrock to be applied.

Of course, a higher-dimension ice flow and hydrological model, over a refined bedrock, would be necessary to go one step further to describe water variability in more detail, but more assumptions concerning the model parameters would also increase model uncertainties. The GF confidence intervals given for the surroundings of Dome C are now about five to ten times lower than those previously available. In the context of the oldest-ice project, we suggest that $\hat{\Phi}_g^m + 2\sigma_\Phi$ is a realistic upper boundary for the local GF value.

### 6.3 Lessons drawn for the oldest-ice research

#### 510 6.3.1 Interpretation of the wet/dry pattern at the base of the ice

The map of basal melt rate (averaged over the last $400\,000$ years) suggests that the old-ice candidates are melt-free across the glacial/interglacial periods. Conversely, the northern part of Dome C is generally a thawing area, whereas Zirizzotti et al. (2012) observed no presence of water, despite the local thick ice. In this region, the climatic signal of the last deglaciation did not reach the bedrock

later than in regions of thinner ice (Fig 9, contour lines). Where the basal ice was cold at the LGM, the pressure melting point has not yet been reached since the deglaciation signal has just reached the bed for the last $\sim 2\,000$ years. In places with thinner ice, the deglaciation signal is almost complete; if the basal ice is still cold today, one can infer it was also the case over the whole glacial/interglacial periods. Most of the always-cold areas could simply be detected by considering the present, cold,

state of the basal ice, and how long since the LGM signal reached the bed. But our heat model is a more comprehensive approach, which provides additional information concerning the local value of the GF and the spatial extent of the cold spots.

#### 6.3.2 Old-ice targets

As expected, the regions where the basal ice is assumed to have been cold throughout the glacial/interglacial

periods are the candidate A and the candidates B, C and D. These three last places may have a lower GF than candidate A and could be better places to exclude the possibility of basal melting. However, they are separated from the topographic dome by the Concordia trench. The deep ice that could be drilled at these locations likely crossed the trench, and the stratigraphy may have been affected by this topographically-disturbed region. Furthermore, the ice velocity is probably higher than over

candidate A because of the steeper slope, and this may enhance basal disturbance. Given the confidence intervals, difference in GF at candidate A is not significant enough to make candidates B, C and D first choices, but they remain spots of great interest. Three-dimensional modelling will now be performed to study the regional ice flow towards these sites in more detail, using the basal melt rate computed in this study as an important input parameter.

Our study also suggests that a certain spot in the northern part of Dome C have a low enough GF to prevent basal melt at long time scales. As the ice is significantly thicker than at other previously considered candidate sites, very old ice could be retrieved with a much better resolution than elsewhere. Given that this hint of a low GF value is the result of only one observation, we wish to emphasize the importance of carrying out additional survey (e.g. ground/airborne radar) to check the

validity of this suggestion.

Finally, the evolution of the benthic $\delta^{18}O$ past sea level proxy (Lisiecki and Raymo, 2005), and its similarity to the one of the EDC reconstructed temperatures, show that the mean air temperature at Dome C was probably $\sim 2\,K$ higher before $-800\,ka$ than after. As a consequence, the mean

basal melt rate was probably also $0.1\,\mathrm{mm\,a^{-1}}$ higher. Even if we conclude this study with favorable
statements, it should be borne in mind that basal ice perhaps underwent some melting on the tops
and flanks of bed reliefs during the Mid-Pleistocene.

## 7    Conclusions

The geothermal heat flux is a poorly constrained geophysical parameter in Antarctica, despite its cru-
cial influence on ice flow properties and old-ice archiving. Around Dome C, the available continental
estimations are currently of limited benefit, and a more precise local estimation is lacking. Here we
present a simple inverse method based on a 1D heat model, constrained by a previous amplitude
analysis of RES echoes recorded at the ice/bedrock interface trying to distinguish wet and dry areas
on the bed. Assuming that the GF is locally uniform, the presence of basal water is only a function
of ice thickness. The critical ice thickness, for which the pressure melting point is reached today, is
inferred from wet/dry thresholds used in the analysis of RES amplitude data (Zirizzotti et al., 2012).
The heat model accounting for the whole history of the ice (changes in temperature, in accumulation
and in ice thickness), was inverted for this critical thickness, to retrieve the value of the GF and of
the time-averaged basal melt rate for ten spots around Dome C.

Our method is valid in dome areas where horizontal advection and diffusion can be neglected, but
its principle could also be adapted for other regions with a more complex physical model. However,
it assumes that the origin of the observed basal water is local, and is thus better suited for flat regions,
where there is no upwards water transport on the bed reliefs. In places where horizontal ice flow is
significant, deformational heat should be taken into account in the energy balance of the ice.

Furthermore, we show that the ice thickness plays a dual role. Of course, on average, it limits the
diffusion heat towards the surface and increases basal melting. But in a changing climate, it lags
behind temperature forcing, so that the base of a thin layer of ice is more sensitive to climate change,
while the base of a thick layer of ice has just begun to be concerned by deglaciation and may still
be cold today. Hence, as the LGM was one of the coldest climatic conditions ever recorded, the lag
effect of the ice thickness must be taken into account to correctly interpret basal conditions today.

The interpolated map of the GF shows a regional gradient, oriented north-south. Where no critical
thicknesses have been measured, the GF values are consistent with the pattern of dry and wet points,
particularly in the northern region, which appears to be dry today, and hosts one potential old-ice
site. All the previously considered old-ice candidates are very likely cold-based, or to have undergone
very little basal melting. The uncertainty at the old-ice targets on the local GF is dramatically reduced
compared to previous estimations made at a continental scale. Our model reveal spatial variations
in the basal melt rate in the Dome C region, which is a helpful and realistic boundary condition
for future 3D ice-flow modelling. More specifically, over the candidate A site, a recent steady sate
model assimilated radar isochrones to invert the value of $\Phi_g$ and $p$, and computed the basal age of

the ice (Parrenin et al., 2017). It confirmed the melt-free areas, giving us confidence that ice as old
as 1.5 million year can be retrieved near Dome C.

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

**Table 1.** Physical parameters, observed (critical thickness $H_c \pm 1\sigma$ [m], ratio of the local surface slope to regional surface slope (ICESat)) and inverted (GHF $\Phi_g \pm 1\sigma_\Phi$ [mW m$^{-2}$]). The surface slopes are computed in circles with a 3-km and 10-km radius centered on each spot, onto which the surface DEM (Bamber et al., 2009) is fitted by a biquadratic function. The ratio between the two slopes is presented below for each spot. A ratio $> 1$ corresponds to a slope locally steeper than its environment.

| Spot | $H_c$ | $\Phi_g^i \pm \sigma_\Phi^i$ | Surf. slope ratio |
|------|-------|------------------------------|-------------------|
| B9 | $3157 \pm 46$ | $55.5 \pm 0.9$ | 1.03 |
| C3 | $2926 \pm 75$ | $59.8 \pm 1.5$ | 1.20 |
| C6 | $2957 \pm 111$ | $59.3 \pm 2.2$ | 1.09 |
| E4 | $3181 \pm 46$ | $55.2 \pm 0.9$ | 0.97 |
| E6 | $3124 \pm 17$ | $56.1 \pm 0.5$ | 2.01 |
| H1 | $3249 \pm 197$ | $53.9 \pm 3.3$ | 1.80 |
| L4 | $3319 \pm 209$ | $53.2 \pm 2.9$ | 1.37 |
| L7 | $3408 \pm 48$ | $51.6 \pm 0.8$ | 1.46 |
| N4 | $3662 \pm 92$ | $48.1 \pm 1.2$ | 1.27 |
| N8 | $3698 \pm 49$ | $47.6 \pm 0.7$ | 1.42 |

**Table 2.** Sensitivity of the GF (assessed from inverse runs, [mW m$^{-2}$]) and basal melt rate (calculated with forward runs, [mm a$^{-1}$]) to input parameters $\alpha$, $\beta$ and $a$. As the final value of $m$ depends on both the inverse run to determine $\Phi_g$, and the forward run to compute the melt rate, the last column accounts for the sensitivity on the whole procedure (inverse+forward).

| Parameter | $\Phi_g$ | $m$ | Total on $m$ |
|-----------|----------|-----|--------------|
| $\alpha = 0.13$ | $-1.4$ | $+0.1$ | $-0.04$ |
| $\alpha = 0.20$ | $+1.4$ | $+0.1$ | $+0.04$ |
| $\beta = 0.138$ | $+0.3$ | $-0.01$ | $+0.02$ |
| $\beta = 0.175$ | $-0.3$ | $+0.01$ | $-0.02$ |
| $a : -10\,\%$ | $-0.3$ | $+0.07$ | $+0.05$ |
| $a : +10\,\%$ | $+0.3$ | $-0.07$ | $-0.05$ |

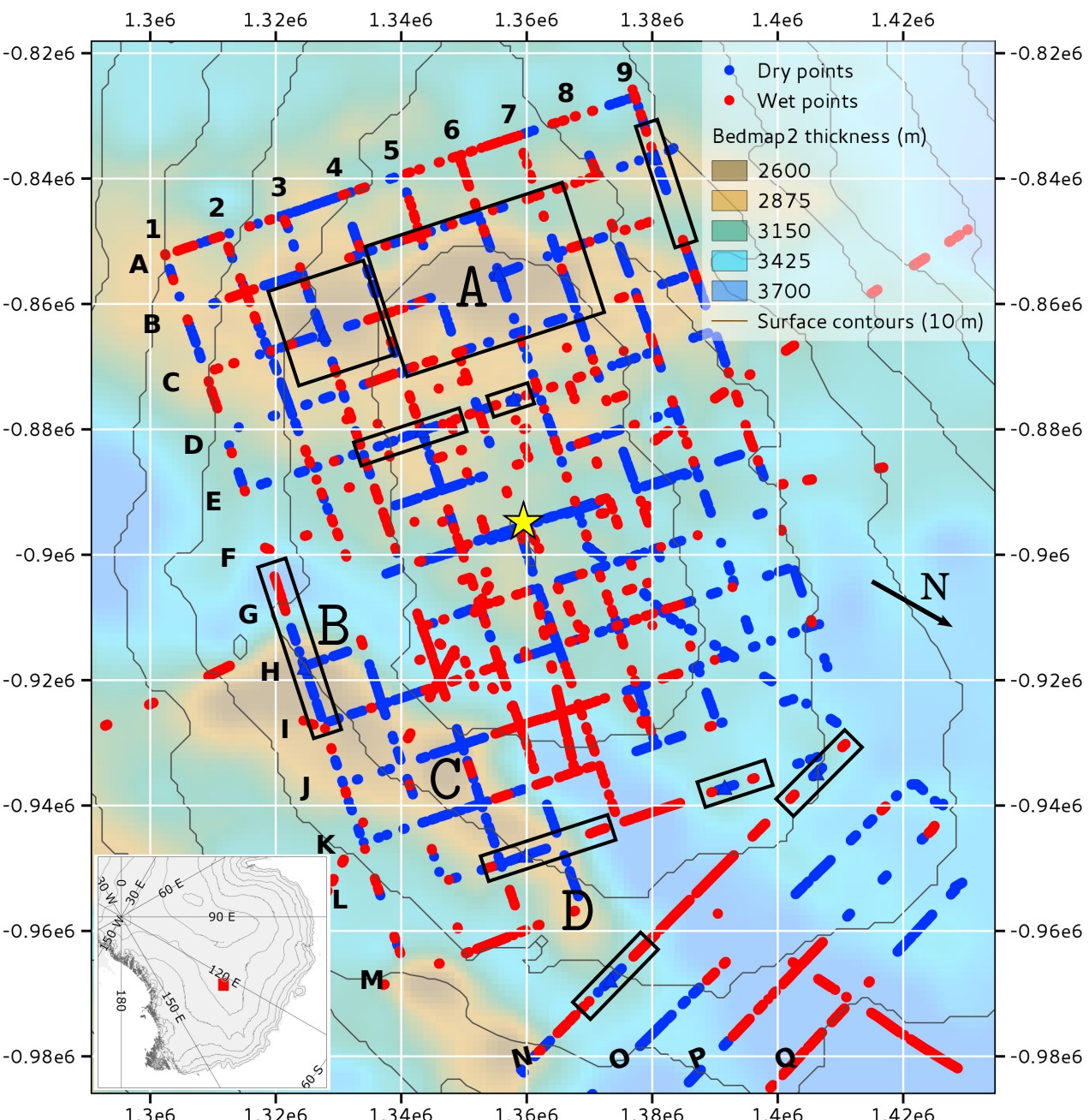

**Figure 1.** Wet (blue) and dry (dry), adapted from Zirizzotti et al. (2012). The yellow star shows the EPICA drill site, located at a distance of $1.4\,\mathrm{km}$ from the topographic dome. Side numbers and letters locate intersection points on the radar grid. Large letters identify for the main sites of interest (candidates A, B, C and D). Projection: WGS84/Antarctic Polar Stereographic - EPSG:3031 (metres). The north direction is bottom right. The situation map on the bottom left shows the position of our domain (red rectangle).

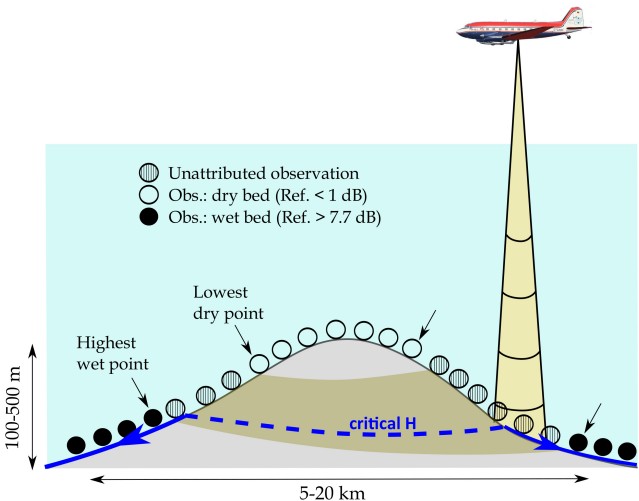

**Figure 2.** This figure illustrates how water and basal topography are linked with a region of flat surface for a uniform local GHF. The blue line shows the presence of water at the bed, and the direction of its flow. The brown area corresponds to a strip where the thermal conditions are unknown. We defined the critical thickness as the minimum thickness that allows basal melting today.

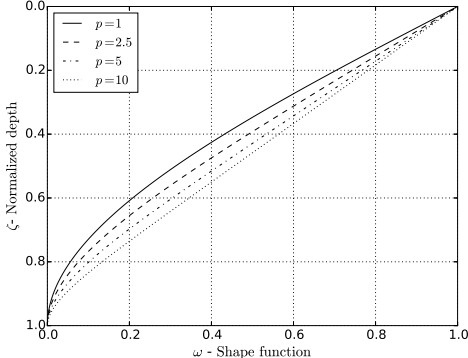

**Figure 3.** Shape function $\omega(\zeta)$ for several values of $p$ (no basal melting).

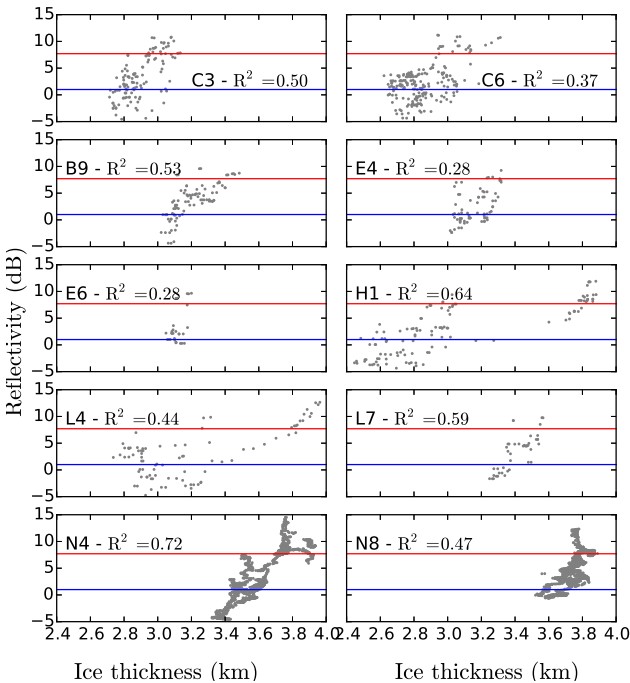

**Figure 4.** Basal reflectivity (dB) vs. ice thickness (km), at 10 favourable spots. Red and blue lines correspond to the thresholds of wet and dry points.

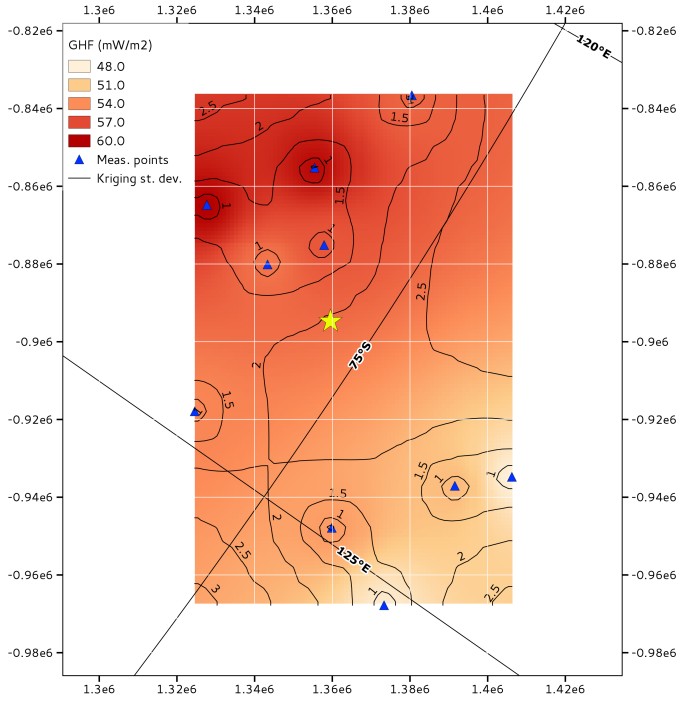

**Figure 5.** Geothermal heat flux, interpolated between the spots (blue triangles), and kriging standard deviation.

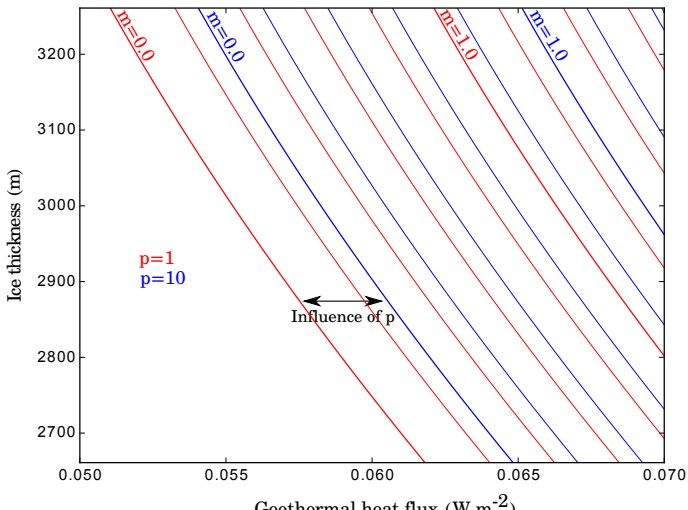

**Figure 6.** Basal melt rate in $\mathrm{mm\,a^{-1}}$, depending on the ice thickness, geothermal heat flux and parameter $p$.

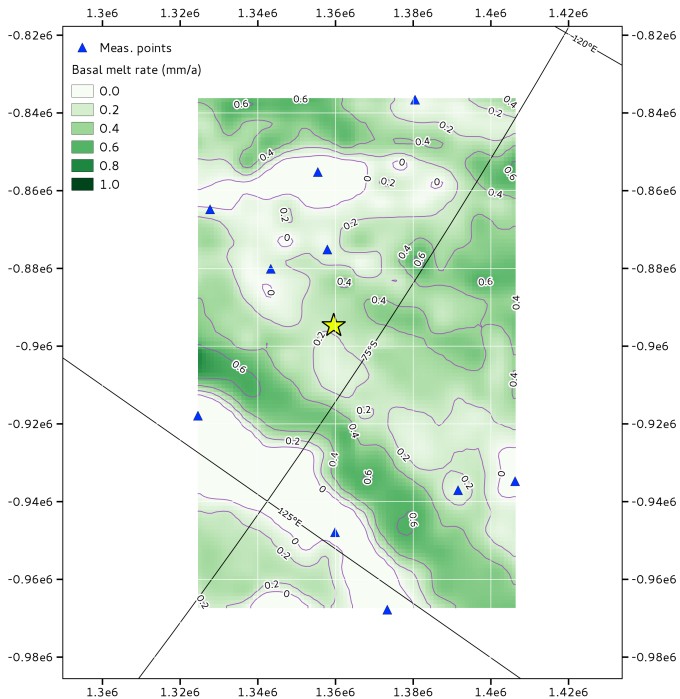

**Figure 7.** Averaged past basal melt rate in $\mathrm{mm\,a^{-1}}$, inferred from the emulating polynomial function with central values of $\Phi_g$ and $p'$. The ice thickness DEM is taken from the Bedmap 2 dataset (Fretwell and coauthors, 2013).

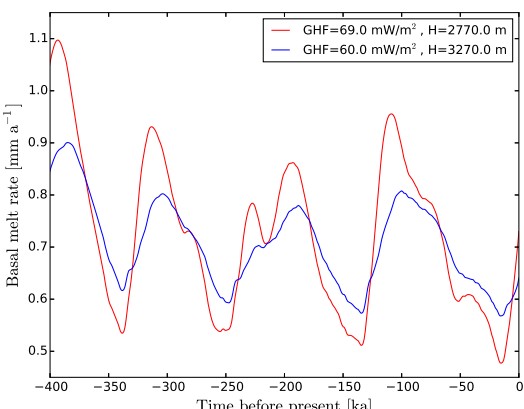

**Figure 8.** Changes in the basal melt rate, depending on the ice thickness and on the geothermal heat flux.

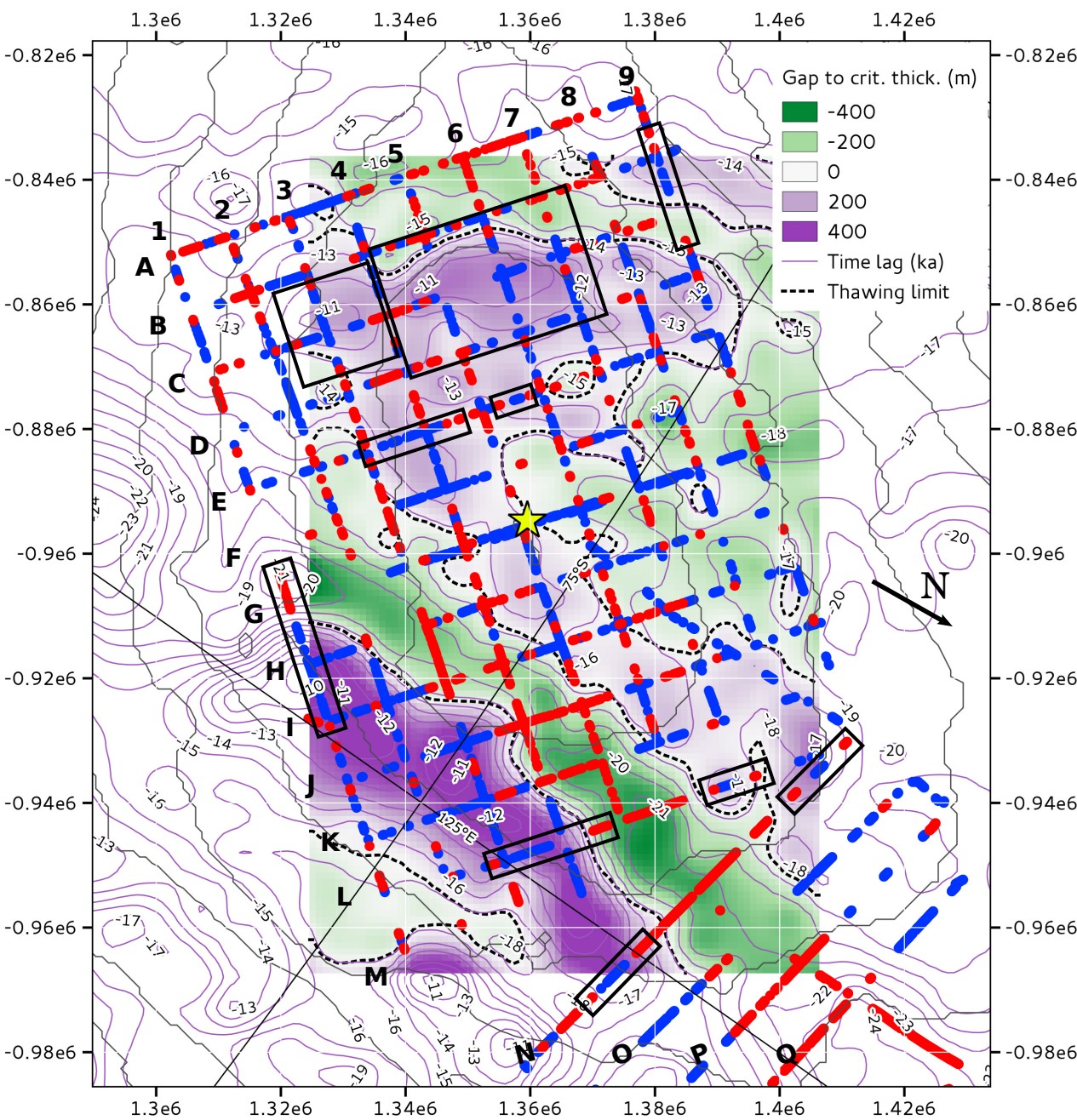

**Figure 9.** Difference between emulated critical ice thickness and observed ice thickness, for $p = 2$ and $\Phi_g = \hat{\Phi}_g^m - 1\,\mathrm{mW\,m^{-2}}$. The ice thickness DEM is taken from the Bedmap 2 dataset (Fretwell and coauthors, 2013). Isocontours show the time lag before the climatic signal to reach the bed.