# Peer review of "Geothermal flux and basal melt rate in the Dome C region inferred from radar reflectivity and heat modelling"

_The Cryosphere, 2017_

## Referee Comment (RC1) · Anonymous Referee #1 · 1 Apr 2017

REVIEW OF "GEOTHERMAL HEAT FLUX AND BASAL MELT RATE IN THE DOME C REGION INFERRED FROM RADAR REFLECTIVITY AND THERMAL MODELLING " By Passalacqua et al., submitted to TCD.

SUMMARY
The paper presents a new method for estimating the geothermal heat flux in regions of slow flowing ice. The authors apply the method to the Dome C region. The study makes use of a combination of radar data to infer wet/dry conditions, the one-dimensional heat equation and inverse methods. The authors first construct a time-dependent, one-dimensional heat model including vertical advection of ice. The model is forced

with past temperature and accumulation rates reconstructed from a deep ice core. The geothermal heat flux is initially assessed at ten spots, where the bed is known to change from wet to dry conditions from radar measurements. The geothermal heat flux is estimated by calculating a critical ice thickness necessary for basal melt, and then applying an inverse method to get the most likely geothermal heat flux. These values are interpolated to the entire region. The heat flux field is then used to calculate melt rates and the authors then arrive at a parameterisation for the melt rate that depend on geothermal heat flux, ice thickness and ice-flow parameter p. This parameterisation is used to calculate the melt rate over time in the region.

MAIN CONCERNS
Overall, the scientific method is sound and it is a nice combination of radar observations, simple ice-flow assumptions and inverse methods. The use of rational assumptions such as negligible variation of the geothermal heat flux on small spatial scales is a good example of how a complicated and under-measured parameter may be simplified. However, I found the structure of the manuscript rather confusing to a point where it detracts from the scientific content. I have listed some of my main points of concern below but overall the manuscript would greatly benefit from a critical revision by the authors with regards to structure, presentation and grammar.

1. Introduction: It is never explicitly mentioned what "old-ice" is. I assume it refers to the on-going international effort of locating ice that is more than 1.2mio years old but the manuscript does not state this nor is the reader told why this is important. Instead the introduction jumps between general statements about geothermal heat and radar data processing, and specific descriptions of a dataset from the region. It is only at the end of p. 3 that the reader is told what the aim of the study is. I suggest splitting the introduction into three sections: i) A general introduction to why "old ice" is important etc. including the general effect of geothermal heat flux and ice thickness on basal melting, ii) an overview of past studies of radar data processing and what have been achieved so far with this technique, and

iii) an introduction to the study region and the specific dataset that this study uses. The authors are of course free to use a different structure but I strongly recommend rewriting the introduction in one way or another. Finally, a figure with a context map would be very helpful. For example, the introduction mentions studies from Thwaites Glacier, West Antarctica, but the Dome C region is in the central part of East Antarctica. A map could prevent confusion as to why the geothermal heat flux values differ significantly between the two sites.

2. Heat model: This section consists of several short subsections that not always follow each other in a logical order. For example, the one-dimensionality of the heat equation is presented first as a model assumption in section 2.1.1., then expanded on in section 2.2 and the reader is presented with the equations in 2.3 and 2.4, while the values of the parameters in the equations are mentioned in sections 2.5 and 2.6. It would greatly improve the readability of this section if the heat model is described first in its entirety, then the velocity model and then the assumptions about geothermal heat flux and water circulation.

3. Basal melt rate emulator: What is the advantage of the "emulator" (I assume this is the same as a parameterisation)? The model is run for the whole domain over the period of 800kyr so why is the parameterisation needed? Can't the model be applied directly to the different scenarios? Is it too computationally expensive?

4. Discussion: Again, I find the order of the sections confusing. The model assumptions and sensitivity tests are followed by a comparison to other studies and then a discussion of the geothermal heat pattern followed by a section titled "Interpretation" (interpretation of what?). I suggest having a separate section with comparison between this study and previous studies, then the discussion section that could start with the overall interpretation and then the discussion of sensitivity tests etc. in the context of the interpretation.

MINOR COMMENTS

There are several small typos e.g. "southest", "explicitely", "conlude", "flown" instead of "flowed", "extend" instead of "extent", "additionnal" that need to fixed. In addition, I have the following comments

Line 5-7: there is a word missing

Line 19: which ice core?

Line 56: What are internal layers? Presumably radar layers but this need to specified and explained why they can be used.

Line 60: How can the method of Carter et al. be used in this study without using the internal layers?

Line 74: "amplitude difference" – is that the same as the difference in returned signal strength?

Line 105: the heat balanace is assumed to be only vertically dependent.

Line 140: Need a few more details here. What are these tabulated parameters? What is the uncertainty in the method? How well do the results compare to those of a 3D model?

Line 143: Missing a word? Coordinate?

Line 160: what is the physical meaning of the parameter p? Is higher values equivalent to more/less basal sliding? What is appropriate for a dome region.

Line 174: The density of the firn layer from Dome C? or from somewhere else?

Line 180: This paragraph seems to contain some information that is irrelevant.

Lines 191-193: These sentences are very confusing. What is too low for what?

Line 209: Odd to use the value 1/6.04 instead of 1.656.

Line 261: Reference to Monte Carlo method missing (e.g. Tarantola, 2005 "Inverse Problem Theory and Methods for Model Parameter Estimation"). Also, from the description of the inversion it does not sound like a Monte Carlo approach but rather like a search of the parameter space. Is it a random parameter space exploration? And how is this done?

Line 274: This is the first mention of potential drill sites. Why C6 and H1?

Line 285: Where is E4 and E6? Which figure is referred to here?

Line 311-317: There should be a reference to Fig. 7 somewhere in this paragraph. Generally, this paragraph is not very clear. E.g. in line 315: How do you assign values to some of the variables? Equations (15) and (16) do not help the reader nor do the several almost identical symbols for different GHF.

Line 323: Point N8 in Fig. 1.

Line 361-373: This is another paragraph that is not clear. For example, what is meant by "much of the map is quite well assigned"? or "well described". Does this mean that the model agrees with the observations? Line 389: "... realistic transport of cold..." Cold snow?

Line 391-385: How does this affect the conclusions?

Line 414-419: Is the accumulation rate influencing the results? That question is raised by not answered in this paragraph.

Line 447: Is it truly Occam's razor or just a lack of good quality data inhibiting model validiation?

Line 482: What clue? Please clarify.

Line 495: The amplitude analysis was not performed in this study but from the sentence it sounds like it was.

FIGURES:

None of the map have an indication of scale. Presumably the axis are in metres but it is not stated anywhere. Figs. 1 and 9 are very busy and could be split up into several maps. Additionally, The combination of magenta/orange and red/blue in Fig. 9 makes it difficult to read.

Fig. 2 is a very nice schematic of the assumptions on this study.

---

## Referee Comment (RC2) · Anonymous Referee #2 · 10 Apr 2017

SUMMARY / MAIN CONCERNS

The authors do a nice job leveraging available data sets and modeling capabilities to provide constraints on the basal melt rate near Dome C. The methods applied seem reasonable and the results seem to advance how to choose a site that is most likely to have preserved more than million-year-old ice. While this is good work, the way the manuscript is written makes it hard to follow, and I think that it does not communicate results in a way that they can be readily used in the broader community effort in the search for an "oldest" ice site. I hope that the authors can both restructure and reword much of the paper in order to make this work more accessible. I do not know exactly how this should be done but will provide some suggestions and identify as many of the

Interactive
comment

grammar mistakes and typos that I can – unfortunately there were many. I recommend starting with reordering and reconfiguring the sections to make sure that each header offers something logical to understand the work, and that text in each section is complete. I recommend that all the authors carefully review the revised manuscript before publication.

I strongly recommend that the authors change the terminology from "geothermal heat flux" to "heat flux" or "geothermal flux"; geothermal heat flux is redundant. (In the same way that 'thermal heat model' would be redundant.) I know the chosen term is often used but hope that stating what is meant more directly will also help the flow of the paper since the term is used throughout the paper.

Introduction to heat-flux estimates from Antarctica doesn't differentiate between east and west, which is confusing the way it is written.

Section 2.1.2 header is misleading when referring to "water circulation", and I am not sure that it is only finding a new word that is needed – is there really a chance of a significant basal hydrological system here? If so, even if a basal spot has not water evident is there a way to constrain that water was not routed through this region?

I wonder if "emulator" is the right word. If you keep it, make sure it is really clear at the start of section where this is introduced what you are doing. Since you are trying to simply approximate the solution to physical equations maybe "approximation" or "parameterization" is a better word?

If you can find a location that has experienced no basal melting that is obviously ideal, but it seems like some melting at some time in the past million years may have occurred and what you really want to make sure is that very old ice is still at the bottom. If there was no melting how old could the deepest ice be? And, for continuity of the record is it critical that no melting has occurred? There needs to be more context on how your results inform the search for very-old ice and how good your results have to be before picking a site. Only at the end it is stated that this work may inform other modeling
efforts – I might have missed it but this should be stated more clearly up front. It would really help to put this work in better context with the larger effort to find a drill site. Of course we won't know what is there until we drill, but I didn't know if your results have really helped to target a site or provided additional information that has to be weighed with all else.

Is the kriged product what would be used in follow-on 3-D modeling? Is this good enough? It seems like the validation you can do for point locations doesn't hold across this whole area in a kriged result, but maybe this interpolated map is the product you need to provide? That wasn't very clear.

If there was more radar data could you do this a lot better? Where is the community at with respect to drilling for oldest ice – will more data be available that you can use in a follow-on study? As a general approach it seems that you already have a lot more data then you might have compared to most places. But, is more needed to really pick the best drill site?

Is there no chance for accretion? Or, have you already ruled out spots where that may have occurred?

MINOR COMMENTS

Line 5: "climate forcing lagged by thick ice" – you mean lag in heat transfer through thick ice, right?

"Temperate" isn't the wrong word, but maybe you want to keep the distinction between "frozen" and "melting"?

Line 29: While the measurement at subglacial Lake Whillans is an important one, I think that it requires some assumptions about the thermal stratification of the lake in order to back out heat flux. I am not an expert in this area but more generally I would make sure that this section fits with your paper. How is this estimate from west Antarctica relevant to Dome C?

Line 37: Statement is about measurements but aren't the references to modeling efforts?

Line 49: should be "increase" instead of "increasing"

I would suggest using "infer", instead of deduce (this occurred in multiple places)

Lines 51-54: I didn't understand this sentence that talks about water routing models – not only was I unsure about how it related scientifically but I'm just not sure what point was being expressed. Perhaps a problem is also that it was a long sentence broken up by citations.

Line 54: Who is "They"? I don't know what study is referred to

Line 57: "significative" should be "significant"

Line 57: I am weary about making a point that uncertainties at a given level are a significant improvement – how uncertain are these uncertainty estimates âŸž

Line 58: I would restate to be something like ". . . but is still too large when trying to find a location that may preserve very old ice"

Line 60-62: Doesn't add much to the reader, either be specific about how what you are doing is or is not similar to previous work or I suggest taking it out

Line 67: Suggest finding another word than "triggered"

I'm not sure it is clear here what you mean by "interesting" – state directly that the glaciology (not glaciologist) community is interested in sites where it has remained cold at the base without melting

Line 88: Again, I would state clearly what you are trying to do and somehow ". . .this paper primarily aims to assess the risk of past temperate conditions at places known to be cold today" doesn't quite get it across. Perhaps something more like "This work constrains sites where basal melting is less likely to have occurred over the past 800

ka, even if they are frozen today, so that very old ice has the best chance of being preserved at that site"

Line 90-94: A little bit of introduction isn't always helpful if the reader doesn't really have a good sense of what is coming. Consider expanding this to be more direct about what it means to do this in "forward mode" and "inverse model"

It may be more clear to talk about this as "running a forward model" and "solving an inverse problem"?

Be consistent with "heat model" or "thermal model" – used interchangeably in title, section headings, and text – but since they mean the same thing just pick one

Line 96: I think you mean "relationships" instead of "relations"

Line 104: "Characteristic" is more often used than "typical"

Line 117: "flown" should be "flowed", or better yet "been transported by ice flow"

Line 121: "permanently" seems too strong

Section 2.1.3 doesn't add much – why is this a section and why does this get mentioned in this order? It seems like you want to introduce the model equations first, then talk about caveats!

Line 138: typo should be "which"

Again, "emulates" seems like the wrong word. Here, maybe "simulates" or "approximates"?

Line 145: Is D dimensionless? Is K a function of reduced depth? (don't think so)

Equation 2: These might be equivalent or just defined differently, but this equation is not the same as in Parrenin et al. (2007; equation 3) – check to make sure no typos between $\zeta$ and $(1\text{-}\zeta)$

Section 2.6 header – suggest stating as "Basal boundary conditions"

Line 87: Why is this an unusual choice?

Line 202: paleotemperatures aren't "known", they are still estimates

Line 209: Why is this represented as 1/6.04K – if report this way make sure to have typed as actual fraction since as it is now with inline slash it could be confusing

Section 2.8 – Do you concentrate grid cells near the base, and is vertical spacing good enough? Also, are 1000 year timesteps good enough? Can you say anything about uncertainties related to your solution grid? The current statement mentions there was a tradeoff between accuracy and speed, how much was compromised?

Everywhere you use "till" should be changed to "until"

Section 3 heading is a bit confusing since it isn't yet clear what is being measured, and it is definitely not heat flux! I would change that to something more clearly related to your data analysis. In general, I suggest coming up with another term for "measurement spots".

Suggest in the text to remind readers what p is. Same for any variable that was introduced awhile back in the text

Line 269: Is there a short justification for the heat flux range of 40-70 mW/m^2?

Line 272: Suggest that "inferred" is better term than "derived"

Line 286: "do not mismatch" – do not match? Or, mismatch?

Line 299: I would put your estimated relationship between heat flux and ice thickness in words and remind the reader that this is for the explored range of heat flux values, or does it hold for any heat flux (any melt rate)? Then, the relationship between heat flux and melt rate must correspond to a specific ice thickness, right?

I lost the details around equations 15-16 and suggest adding more text to explain what this means and how it was derived.

[Figure]

Line 337: what do you mean by "at the opposite" here?

Line 351: Again, not sure how to take this point about an "unusual choice" and why it was chosen that way to begin with

Line 354: Why is "order of magnitude" good enough?

Line 357: The leading sentence doesn't seem to relate to the second sentence and I had to read back to see if I understood what was coming and has already been done. Suggest better lead-in to this section.

Line 365: "build" should be "built", and really do you mean "calculated"?

Line 368: What small-scale structures are these E, G, H, I, L locations referring to?

Line 369: Should be "non-melting"

Line 371: Need to rephrase part about "... is however often respected ..."

"undoubtly" should be "undoubtedly"

Line 373: What do you mean by "local gap"?

Line 375: Should be "dependent"

Section 6.1, maybe "Method validation"? There is no lead in, which may be fine but as is now it isn't clear how come you need to have overarching section 6.1

Line 399: What do you mean "the surface slope is the source of motion"?

Talking about clues isn't very precise – is there a better word?

Paragraph around lines 415-420: Not sure I understood the point of this paragraph, if this is just too hard to constrain state that directly. If it needs to be considered somehow and will significantly affect uncertainty estimate state that directly too. I can't tell if this is something that really could matters.

Line 422: Not sure what you mean with "litmus"

Section 6.1.4 – by "structure" do you mean "spatial variation"

FIGURES AND TABLES

Would be worth defining input parameters in Table 2. I don't quite understand what "total on m" represents.

Figure 2: I might have missed it but check that critical ice thickness is defined in the text to this point, or add to caption

Figure 4: Not sure it was discussed in the text how the 10 spots were chosen?

Figure 6: A lot of overlapping lines. How many discrete values of m are represented?

---

## Short Comment (SC1) · 11 Apr 2017

Might be good to compare the results of this interesting analysis with the work I did with Julian Dowdeswell in 1996:

Siegert, M.J. & Dowdeswell, J.A. Spatial variations in heat at the base of the Antarctic Ice Sheet from analysis of the thermal regime above sub-glacial lakes. Journal of Glaciology, 42, 501-509. (1996).

See attached.

The values are quite similar I think.

Martin

Please also note the supplement to this comment:
http://www.the-cryosphere-discuss.net/tc-2017-23/tc-2017-23-SC1-supplement.pdf

---

## Author Comment (AC1) · 19 Jun 2017

The paper presents a new method for estimating the geothermal heat flux in regions of slow flowing ice. The authors apply the method to the Dome C region. The study makes use of a combination of radar data to infer wet/dry conditions, the one-dimensional heat equation and inverse methods. The authors first construct a time-dependent, one-dimensional heat model including vertical advection of ice. The model is forced with past temperature and accumulation rates reconstructed from a deep ice core. The geothermal heat flux is initially assessed at ten spots, where the bed is known to change from wet to dry conditions from radar measurements. The geothermal heat flux

is estimated by calculating a critical ice thickness necessary for basal melt, and then applying an inverse method to get the most likely geothermal heat flux. These values are interpolated to the entire region. The heat flux field is then used to calculate melt rates and the authors then arrive at a parameterisation for the melt rate that depend on geothermal heat flux, ice thickness and ice-flow parameter p. This parameterisation is used to calculate the melt rate over time in the region.

MAIN CONCERNS Overall, the scientific method is sound and it is a nice combination of radar observa- tions, simple ice-flow assumptions and inverse methods. The use of rational assump- tions such as negligible variation of the geothermal heat flux on small spatial scales is a good example of how a complicated and under-measured parameter may be simpli- fied. However, I found the structure of the manuscript rather confusing to a point where it detracts from the scientific content. I have listed some of my main points of concern below but overall the manuscript would greatly benefit from a critical revision by the authors with regards to structure, presentation and grammar.

We would like to thank the reviewer for the fruitful comments made on our work. We agree that the structure of certain sections needed to be modified to make the article more clear-cut. In particular, the introduction section now benefits from a better explained 'oldest ice' context, and section 2 (method) and 6 (discussion) were reordered. We also agree with almost all the minor comments and tried to correct the text accordingly.

The present version of the paper has been corrected by a native speaker for english wording.

When needed, the old sentences are colored in green with corresponding line, the new ones in red.

1. Introduction: It is never explicitly mentioned what "old-ice" is. I assume it refers to the on-going international effort of locating ice that is more than 1.2mio years old but the manuscript does not state this nor is the reader told why this is important. Instead

the introduction jumps between general statements about geothermal heat and radar data processing, and specific descriptions of a dataset from the region. It is only at the end of p. 3 that the reader is told what the aim of the study is. I suggest splitting the introduction into three sections: i) A general introduction to why "old ice" is important etc. including the general effect of geothermal heat flux and ice thickness on basal melting, ii) an overview of past studies of radar data processing and what have been achieved so far with this technique, and iii) an introduction to the study region and the specific dataset that this study uses. The authors are of course free to use a different structure but I strongly recommend rewriting the introduction in one way or another. Finally, a figure with a context map would be very helpful. For example, the introduction mentions studies from Thwaites Glacier, West Antarctica, but the Dome C region is in the central part of East Antarctica. A map could prevent confusion as to why the geothermal heat flux values differ significantly between the two sites.

We followed the above suggestions, in particular to explain how this work is involved in the more general frame of the oldest-ice research. We also added a configuration map. The introduction is now clearly separated in 3 subsections :

1.1) The oldest ice research

Why oldest ice, and how basal melting can be avoided

1.2) GF assessment methods unders ice sheets

Concerning Thwaites Glacier, we wanted to point out the method and its accuracy, not the particular value of the GF.

1.3) Exploitation of available RES dataset around Dome C

Here we explain how we want to use available RES data, and we set the goal of the paper : L. 107 : "this work aims at constraining sites known to be frozen today and that are very likely to have been frozen in the last 800 ka as well."

2. Heat model: This section consists of several short subsections that not always follow

each other in a logical order. For example, the one-dimensionality of the heat equation is presented first as a model assumption in section 2.1.1., then expanded on in section 2.2 and the reader is presented with the equations in 2.3 and 2.4, while the values of the parameters in the equations are mentioned in sections 2.5 and 2.6. It would greatly improve the readability of this section if the heat model is described first in its entirety, then the velocity model and then the assumptions about geothermal heat flux and water circulation.

We reordered the text as suggested for the "Heat model" section.

2.1 Geometry and coordinate system

2.2 Heat equation

2.2.1 Ice thermal properties

2.2.2 Basal boundary conditions

2.2.3 Boundary condition at the surface

2.3 Velocity model

2.4 Proceeding assumptions

2.4.1 Correspondance between wet and thawing areas

2.4.2 GF spatial variability

3. Basal melt rate emulator: What is the advantage of the "emulator" (I assume this is the same as a parameterisation)? The model is run for the whole domain over the period of 800kyr so why is the parameterisation needed? Can't the model be applied directly to the different scenarios? Is it too computationally expensive?

We understand "emulator" as the process we use to empirically replace the whole heat model, to save computation time. We added this new sentence to make it more clear :

L. 344 : "As the computation for a given set of parameter lasts several minutes, computing the basal melt rate for each point of our domain would be far too expensive. Here, the result of the whole forward model is mimicked by an emulator that depends on the input parameters H, $\Phi_g$ and p"

Giving an emulator will also be useful to help anyone who wants to compute realistic basal melt rate with a future refined bed for example, and starting from the map of GF.

4. Discussion: Again, I find the order of the sections confusing. The model as- sumptions and sensitivity tests are followed by a comparison to other studies and then a discussion of the geothermal heat pattern followed by a section titled "In- terpretation" (interpretation of what?). I suggest having a separate section with comparison between this study and previous studies, then the discussion section that could start with the overall interpretation and then the discussion of sensitivity tests etc. in the context of the interpretation.

Following your suggestions, we reordered the section :

6.1 Consistency with published data and measurements

6.2 Model assessment

6.2.1 Method validity

6.2.2 Sensitivity to parameters

6.2.3 Spatial variation of the GF field

6.3 Lessons drawn for the oldest-ice research

6.3.1 Interpretation of the wet/dry pattern at the ice base

6.3.2 Old-ice targets

MINOR COMMENTS There are several small typos e.g. "southest", "explicitely", "conlude", "flown" instead of "flowed", "extend" instead of "extent", "additionnal" that need to fixed. In addition, I have the following comments Line 5-7: there is a word missing

The new sentences are the folowing :

"But, since basal conditions depend on heat transfer forced by climate but lagged by the thick ice, the basal ice may currently be frozen whereas in the past it was generally melting. For that reason, the risk of bias between present and past conditions has to be evaluated."=

Line 19: which ice core?

The EPICA Dome C ice core (EDC) is now explicitly mentionned.

Line 56: What are internal layers? Presumably radar layers but this need to specified and explained why they can be used.

L 56 : "basal melt rates have been estimated by fitting the vertical strains with dated internal layers"

Radar layers are used to constrain the vertical advection of ice, which is one of the component of the energy budget. The sentence is now :

L.80 : "basal melt rates for the region north of Dome C have been estimated by fitting the vertical strain rates with dated radar layers, to constrain the vertical advection of ice, and its energy budget"

Line 60: How can the method of Carter et al. be used in this study without using the internal layers?

L60 : "Without using internal layers, which are not available everywhere, we will follow a similar approach to these two last studies"

Our formulation was ambiguous, we do not use the method of Carter et al. The sentence is now :

L.85 : "As dated layers are not available everywhere, we will follow a radar-based approach like that of Schroeder et al. (2014), but adapted to the specific pattern of

radar echoes under Dome C"

Line 74: "amplitude difference" – is that the same as the difference in returned signal strength?

We can also talk of radar echo strenght but the calculation of reflectivity done in the paper of Zirizzotti et al (2012) was based on the difference between the amplitude (strenght or power) of the radar wave reflected by the surface and the amplitude of the radar wave reflected from the ice/rock (or water) interface in dBm. We prefer to keep "amplitude difference" for consistency with the 2012 paper.

Line 105: the heat balanace is assumed to be only vertically dependent.

New sentence L134 :"Thus, the heat balance is assumed to be only vertically dependent."

Line 140: Need a few more details here. What are these tabulated parameters? What is the uncertainty in the method? How well do the results compare to those of a 3D model?

L139 : "At each timestep, explicit expressions for the thickness and the bedrock elevation are solved, that depend on the accumulation rate and six tabulated parameters."

The parameters account for the sensitivity of the ice thickness, surface, height, etc. to the climate forcing.We do not go into deeper details here, since the model is fully explained in Parrenin et al 2007, and since the influence of the accumulation and temperature reconstruction, used as forcing, does not deeply change the results. However, we added a few explanations :

L.143 : "... six tabulated parameters, that account for the sensitivity of physical quantities (in particular bedrock and surface height and ice thickness) to the climate forcing."

Line 143: Missing a word? Coordinate?

Yes, "coordinate was missing : L.148 : "The heat balance of ice only depends on the

vertical coordinate"

Line 160: what is the physical meaning of the parameter p? Is higher values equivalent to more/less basal sliding? What is appropriate for a dome region.

We completed the paragraph with additionnal explanations concerning the construction of synthetic profiles :

L.217 : "Far from divides, and for an isotropic ice, this parameter depends on the non-linearity of the ice rheology and the vertical temperature gradient at the base (Lliboutry, 1979):

p=n-1+Q/RT2*dT/dz

where n is the exponent of the Glen's flow law, $Q = 60$ kJ mol$^{-1}$ is an activation energy, $R = 8.314$ J mol$^{-1}$K$^{-1}$ the gas constant, and Tb the basal temperature. Following Eq.(11), the values of p should range within 7 to 9 on the East Antarctic plateau. But in practice we will use p close to divides in a larger value range, as a parameter able to account for realistic vertical velocity profiles. For exemple, dome profiles are expected to correspond to low p values due to Raymond arches (Raymond, 1983), whereas basal sliding will make the profile more linear and increase the value of p."

Line 174: The density of the firn layer from Dome C? or from somewhere else?

L.164 : "The density profile of the Dome C firn layer is taken from Parrenin et al. (2007)"

Line 180: This paragraph seems to contain some information that is irrelevant.

We wanted to be precise on our choice concerning the expression of the pressure melting point. But we understand these explanations may make the reader lose his train of thoughts, so now we just say what is our expression :

L.168 : "For thawing glacier ice, the melting temperature T m linearly depends on the ice pressure P and the partial pressure of air dissolved in the ice P 0 , which is expressed as (Ritz, 1992): ..."

Lines 191-193: These sentences are very confusing. What is too low for what?

L. 192 : "The melting temperature computed with B = 0.098 K.Pa -1 would be 0.8 K too low, whereas, it is found to be 270.96 K with Eq.(9)"

The sentence is reworded, just saying that the equation is close to the observation :

L173 : "The temperature profile can be extrapolated to the bedrock (similar to Dahl-Jensen et al. (2003) at North GRIP) to 271.04 K, where Eq. (5) finds 270.96 K."

Line 209: Odd to use the value 1/6.04 instead of 1.656.

We prefer to keep it like this for consistency with literature (Lorius and Merlivat, 1975).

Line 261: Reference to Monte Carlo method missing (e.g. Tarantola, 2005 "Inverse Problem Theory and Methods for Model Parameter Estimation"). Also, from the description of the inversion it does not sound like a Monte Carlo approach but rather like a search of the parameter space. Is it a random parameter space exploration? And how is this done?

The parameter space is not explored in a uniform way, because each parameter couple (H, p) is given random values along a gaussian joint-distribution. So we think "Monte-Carlo" is the appropriate word for what we did. The gaussian distribution for H and p are described in the rest of the section.

Line 274: This is the first mention of potential drill sites. Why C6 and H1?

Now a specific sentence introduces the denomination of the points in the introcution section and figure 1 is referred.

L.121 : "For convenience, the domain is referenced with letter-figure couples, corresponding to the grid of the Italian survey (Fig. 1). In particular, two promising old-ice candidates are located at C6 and H1."

Line 285: Where is E4 and E6? Which figure is referred to here?

The previous sentence (L 121) now gives the needed information as well.

Line 311-317: There should be a reference to Fig. 7 somewhere in this paragraph. Generally, this paragraph is not very clear. E.g. in line 315: How do you assign values to some of the variables? Equations (15) and (16) do not help the reader nor do the several almost identical symbols for different GHF.

This paragraph is taken off, since it was unnecessarily complicated with respect to what we want to show. Now we simply present the past basal melt rate calculated with the central values of GF and p, which is the main message of the paper.

Line 323: Point N8 in Fig. 1. Reference to figure 1 added L. 371

Line 361-373: This is another paragraph that is not clear. For example, what is meant by "much of the map is quite well assigned"? or "well described". Does this mean that the model agrees with the observations? Line 389: ". . . realistic transport of cold..." Cold snow?

We agree that this paragraph was not precise enough, and it was reworded to make it more clear-cut.

L.414 : "Superimposed on the observed data, the model output shows that large-scale patterns of wet-dry areas are respected, especially on steep bed slopes (candidate B, C, D, and to a lesser extent candidate A). On these bed reliefs, certain points however show a discrepancy between model and observation, but the gap to the critical thickness is often close to 0 m (D3, D5, D8, M3), meaning that a small change in GF forcing, or a better description of the ice thickness, would better assign these particular points. The 1 km-resolution of the Bedmap 2 bedrock dataset (Fretwell and coauthors, 2013) smoothed along-track subkilometric features detected by our RES survey.

The steeper the bed, the sharper the limit between melting and non-melting areas. In the central, flatter, part of the domain, the position of this limit is more blurred. As we could not assess the GF with our method there, it was interpolated. Despite this lack of

constraints, several small-scale features are well mimicked (dry areas at I9, G-H8, wet areas at G9 and L7). Other regions were not attributed in agreement with observations (G6-7-8, H-I8), meaning that the GF is overestimated, probably up to 3mW m-2 , which is consistent with the uncertainties given by our method (inversion and interpolation)."

Line 391-385: How does this affect the conclusions?

L 391 : "Some of the given confidence intervals are quite low (E4, E6, L7 and N8), and this is a consequence of the tiny altitude difference measured on Fig. 1 between the highest wet points and thelowest dry ones for a given spot. Since the correlation between ice thickness and reflectivity was weak, the confidence intervals at E4 and E6 are probably underestimated, and some local effects may not have been accounted for in this study (e.g. small relief and GHF variabilities)."

As the GF values found for these point are not strongly discordant with the closer points, no particular warning should be placed. But we had to mention these points because we know the observation constraint is probably not very good there (now this pargraph is at L.467)

Line 414-419: Is the accumulation rate influencing the results? That question is raised by not answered in this paragraph.

L. 487 : "The sensitivity of the GHF on the surface accumulation is less than a few tenths of mW m -2 so that, accounting for its spatial variations would not radically modify our results."

Line 447: Is it truly Occam's razor or just a lack of good quality data inhibiting model validiation?

L. 447 : "This application of the parcimony principle of Occam's razor supports the validity of our 1D approach, and means that the main physical mechanisms have been accounted for."

We agree with your remark. We think that our quite simple model works quite well

on bed reliefs, where the results show patterns similar to the observations. On flatter areas, the quality of the method is more difficult to assess, given the quality of the bed DEM, and the compatibility with our assumptions (no upstream flow of basal water). So we reworded the paragraph thiws way :

L. 494 : "This means that the main physical mechanisms have been taken into account, at least where it was possible to evaluate the critical thickness on significant topographic features. Where the GF is interpolated, in flatter areas, the lack of constrains prevents us from really assessing the validity of the method, meaning that our method needs a sufficient hilly bedrock to be applied."

Line 482: What clue? Please clarify.

L. 537 : "Given that this hint of a low GF value is the result of only one observation,..."

Line 495: The amplitude analysis was not performed in this study but from the sentence it sounds like it was. L. 550 : "a previous amplitude analysis"

FIGURES: None of the map have an indication of scale. Presumably the axis are in metres but it is not stated anywhere. Figs. 1 and 9 are very busy and could be split up into several maps. Additionally, The combination of magenta/orange and red/blue in Fig. 9 makes it difficult to read.

The projection and coordinate system are specified in the caption of figure 1. Colours of figure 9 were changed.

We could not split the figures 1 and 9, because the information they contain is much richer when superimposed the ones on the others. We hope their large size should be enough for the clarity.

Fig. 2 is a very nice schematic of the assumptions on this study.

---

## Author Comment (AC2) · 19 Jun 2017

The authors do a nice job leveraging available data sets and modeling capabilities to provide constraints on the basal melt rate near Dome C. The methods applied seem reasonable and the results seem to advance how to choose a site that is most likely to have preserved more than million-year-old ice. While this is good work, the way the manuscript is written makes it hard to follow, and I think that it does not communicate results in a way that they can be readily used in the broader community effort in the search for an "oldest" ice site. I hope that the authors can both restructure and reword much of the paper in order to make this work more accessible. I do not know exactly

how this should be done but will provide some suggestions and identify as many of the starting with reordering and reconfiguring the sections to make sure that each header offers something logical to understand the work, and that text in each section is complete. I recommend that all the authors carefully review the revised manuscript before publication.

We would like to thank the reviewer for the constructive comments made on our work. We agree that the structure of certain sections need to be modified to make the article more clear-cut. In particular, the introduction section now benefits from a better explained 'oldest ice' context, and section 2 (method) and 6 (discussion) were reordered. We also agree with almost all the minor comments and tried to correct the text accordingly.

The present version of the paper has been corrected by a native speaker for english wording.

When needed, the old sentences are colored in green with corresponding line, the new ones in red.

I strongly recommend that the authors change the terminology from "geothermal heat flux" to "heat flux" or "geothermal flux"; geothermal heat flux is redundant. (In the same way that 'thermal heat model' would be redundant.) I know the chosen term is often used but hope that stating what is meant more directly will also help the flow of the paper since the term is used throughout the paper.

That makes sense, and we changed the terminology for "geothermal flux".

Introduction to heat-flux estimates from Antarctica doesn't differentiate between east and west, which is confusing the way it is written.

We understand that the text was potentially confusing. Now we focus specifically on the method to assess the GF under ice sheets in general (§1.2 GF assessment methods unders ice sheets), and we specify East or West Antarctica when needed.

Section 2.1.2 header is misleading when referring to "water circulation", and I am not sure that it is only finding a new word that is needed - is there really a chance of a significant basal hydrological system here? If so, even if a basal spot has not water evident is there a way to constrain that water was not routed through this region?

The idea here is to check if wet and dry areas really correspond to thawing and frozen areas (title changed as follows) :

2.4.1 Correspondance between wet and thawing areas

The true basal hydrological system cannot be observed easily and we are limited to theory. If there is no basal water circulation, Zirizotti map is just enough, but this would be a too optimistic assumption : we need to investigate the pessimistic case where dry areas are in fact thawing. Two clues are giving us confidence that dry areas are frozen, as explained below : the continuous nature of the melting, but also the relatively steeper surface slope at the spots (previously discussed in §6.1.1, now in this section) :

"A continuous hydrological network upstream is only fed by local melting, so that, unless the network is disconnected, the thawed water cannot be driven out faster than the melt rate. As a consequence, some water would always remain at the base of the melting ice. Furthermore, water at the base enables basal sliding, and reduces the basal drag. For a given ice flux, the surface slope gives some indication on the relative importance of internal deformation and basal sliding in the ice motion. Local steeper surface may be associated with more basal drag and ice deformation, whereas local flatter surface may be associated with more sliding. Most of the spots where the model will be run are standing on slopes locally steeper than the regional slope"

I wonder if "emulator" is the right word. If you keep it, make sure it is really clear at the start of section where this is introduced what you are doing. Since you are trying to simply approximate the solution to physical equations maybe "approximation" or "parameterization" is a better word?

We added the following sentence to define the word properly and say why we need it. We understand "emulator" as the process we use to empirically replace the whole heat model, to save computation time. We do not intend to parameterize "true" physical relations between input parameters, as we do not specifically account for diffusion or advection for example, but aggregate all the effects in one empirical expression.

L.347 : "As computing a given set of parameter takes several minutes, computing the basal melt rate for each point of the Dome region would be far too expensive. Here, the result of the whole forward model is mimicked by an emulator that depends on the input parameters H, $\Phi_g$ and p"

If you can find a location that has experienced no basal melting that is obviously ideal, but it seems like some melting at some time in the past million years may have occurred and what you really want to make sure is that very old ice is still at the bottom. If there was no melting how old could the deepest ice be? And, for continuity of the record is it critical that no melting has occurred? There needs to be more context on how your results inform the search for very-old ice and how good your results have to be before picking a site. Only at the end it is stated that this work may inform other modeling efforts - I might have missed it but this should be stated more clearly up front. It would really help to put this work in better context with the larger effort to find a drill site. Of course we won't know what is there until we drill, but I didn't know if your results have really helped to target a site or provided additional information that has to be weighed with all else.

Is the kriged product what would be used in follow-on 3-D modeling? Is this good enough? It seems like the validation you can do for point locations doesn't hold across this whole area in a kriged result, but maybe this interpolated map is the product you need to provide? That wasn't very clear.

If a bit of thawing occurred between 1.5M and 1M years ago, and no melting after, the basal ice age is clearly not "infinite", but 1.5M-year ice still exists somewhere, since it

existed 1M years ago. The age resolution must be affected but this kind of question goes beyond the scope of this paper, and mechanical modelling is needed.

We now better set the wider frame of this paper (observations - heat model - mechanical model) in the introduction (1.1 The oldest ice research). As geothermal flux is largely unkonwn, any new information on it is useful in the decision process for the drill site. The 3D-mechanical model we use now tries to constrain the ice rheology ; as it is an inverse process, the informations given on the basal melting (average + confidence intervals) reduce the parameter space to explore. Now, we use these kriged maps as prior values for 3D modelling. As we are focusing on candidate A, where we do have some contrains on the geotharmal flux, the kriged products are good enough there, and they even are probably more realistic than the existing products in the region (almost uniform value).

L. 45 : "In a context of locating the oldest-ice, our objective was to constrain the value of the geothermal flux and of the basal melt rate around Dome C. Later, this information will be used as a boundary condition for 3D mechanical simulations. As 3D models require many input parameters and are computationnaly expensive, any additional information about the thermal regime of the ice that reduces parameters value range is welcome, and will help select a suitable drill site."

If there was more radar data could you do this a lot better? Where is the community at with respect to drilling for oldest ice - will more data be available that you can use in a follow-on study? As a general approach it seems that you already have a lot more data then you might have compared to most places. But, is more needed to really pick the best drill site?

From this method point of view, a better description of the bed topography and bed reflectivity in the central, flatter, part of our study area could bring some interesting constrains. Elsewhere we do not think we could do much better. The present study gives us confidence in the absence of melting over the candidate A, but to constrain the

effective age and age resolution we need additional radar information. Internal layers within the ice are now available over the candidate A (Young et al, TCD, in review and Parrenin et al, TCD, in review). By continuity from the EPICA core, these layers can be dated, so that the age-depth relation is described on the 3/4 of the ice column. This information is mandatory to have a constrain on datation models (1D or 3D), and to know the quality of the stratigraphy.

L577 : "More specifically, over the candidate A, a recent steady sate model assimilates radar isochrone layers to invert the value of $\Phi_g$ and p, and computes the basal age of the ice (Parrenin et al., 2017)."

Is there no chance for accretion? Or, have you already ruled out spots where that may have occurred?

Accretion is possible where the ice goes from wet to dry during interglacials. This could occurr on the flanks of the candidate A. At least on the candidate A, no accretion clue were visible in the last radar survey, so we did not investigate this particular point.

MINOR COMMENTS

Line 5: "climate forcing lagged by thick ice" - you mean lag in heat transfer through thick ice, right?

The sentence is completed as follows :

L. 5 : "But, as the basal conditions depend on heat tranfer forced by climate but lagged by the thick ice, the basal ice may be frozen today..."

"Temperate" isn't the wrong word, but maybe you want to keep the distinction between "frozen" and "melting"?

We agree with your words, that more focus on the melting process that detroy old-ice layers.

L.5 : "the basal ice may be frozen today whereas it was in average melting in the past."

Line 29: While the measurement at subglacial Lake Whillans is an important one, I think that it requires some assumptions about the thermal stratification of the lake in order to back out heat flux. I am not an expert in this area but more generally I would make sure that this section fits with your paper. How is this estimate from west Antarctica relevant to Dome C?

We just wanted to illustrate that measuring the heat flux is not an easy task in Antarctica, so that inverse methods and modelling are needed on the antarctic plateau. But as it was confusing, the text is now more straightforward on East Antarctica.

L. 52 : "The geothermal flux is usually derived from temperature gradients meassured in boreholes in the ground, but this cannot be done easily below ice sheets, because of the difficult access to the bedrock."

Line 37: Statement is about measurements but aren't the references to modeling efforts?

L37 : "Without any available temperature measurements, the value of the GHF has been estimated from geological considerations"

The sentence was confusing, now reworded as follows :

L.58 : "As deep boreholes are not numerous in East Antarctica, the value of the GF has first been estimated from geological considerations"

Line 49: should be "increase" instead of "increasing"

Correction made L. 72

I would suggest using "infer", instead of deduce (this occurred in multiple places)

Correction made

Lines 51-54: I didn't understand this sentence that talks about water routing models - not only was I unsure about how it related scientifically but I'm just not sure what point

was being expressed. Perhaps a problem is also that it was a long sentence broken up by citations.

We added a link between the first and the second part of the paragraph, to make clear why considering basal hydrology is needed :

L.75 : "But the presence of water is not a sufficient clue to infer the GF, since water either comes from local basal melting or was routed from elsewhere. Using a collection of water routing models, Schroeder et al. (2014) inferred the value of the GF needed to explain the observed pattern of radar echoes..."

Line 54: Who is "They"? I don't know what study is referred to

L54 : "They derived an average value of 114 ± 10 mW m -2 for the Thwaites Glacier catchment."

L. 77 : "Schroeder et al. (2014) inferred the value of the GF needed to explain the observed pattern of radar echoes, and derived an average value of 114 ± 10 mW m -2 ..."

Line 57: "significative" should be "significant"

L. 57 : "The uncertainty on the GHF was estimated ±12 mW m -2 , which is a significative improvement"

L. 82 : "The uncertainty on the GF estimation was ±12 mW m -2 , which is a significant improvement"

Line 57: I am weary about making a point that uncertainties at a given level are a significant improvement - how uncertain are these uncertainty estimates

The current available (continental) estimations are affected by large uncertainties of +/- 20 mW/m2 (Fox Maule 2005 for example). From the "oldest-ice" point of view, these estimations cannot constrain efficiently the local basal melt rate. The key information for us would be a realistic upper boundary for the GF that would anyway allow the ice

to be frozen through time, and the only way is to find places where there is no melting. In that sense, any effort to reduce locally the uncertainties of continental estimations are welcome.

Line 58: I would restate to be something like ". . . but is still too large when trying to find a location that may preserve very old ice"

L. 82 : "The uncertainty on the GF estimation was ±12 mW m -2 , which is a significant improvement compared to the previously available estimations, but is still too large when trying to find a location that may preserve very old ice."

Line 60-62: Doesn't add much to the reader, either be specific about how what you are doing is or is not similar to previous work or I suggest taking it out

L58 :"Moreover, their study area does not completely cover the main old-ice candidates (east and south-west of Dome C). Without using internal layers, which are not available everywhere, we will follow a similar approach to these two last studies, but adapted to the specific pattern of radar echoes under Dome C"

The sentence is reworded to say that we need a new local estimation.

L. 84 : "Their study area does not cover the main old-ice candidates located to the east and south-west of Dome C, so a new local estimation is needed. As dated layers are not available everywhere, we will follow a reflectivity-based approach like Schroeder et al. (2014), but adapted to the specific pattern of radar echoes under Dome C"

Line 67: Suggest finding another word than "triggered" I'm not sure it is clear here what you mean by "interesting" - state directly that the glaciology (not glaciologist) community is interested in sites where it has remained cold at the base without melting

This paragraph is taken out, but the main ideas are put in the first paragraph (§1.1 The oldest-ice research, L. 37).

Line 88: Again, I would state clearly what you are trying to do and somehow ". .

.this paper primarily aims to assess the risk of past temperate conditions at places known to be cold today" doesn't quite get it across. Perhaps something more like "This work constrains sites where basal melting is less likely to have occurred over the past 800ka, even if they are frozen today, so that very old ice has the best chance of being preserved at that site"

L 88: "this paper primarily aims to assess the risk of past temperate conditions at places known to be cold today"

We restated this sentence as follows :

L. 107 : "this work aims at constraining sites known to be frozen today and that are very likely to have been frozen in the last 800 ka as well, increasing the probability that very old ice has been preserved."

Line 90-94: A little bit of introduction isn't always helpful if the reader doesn't really have a good sense of what is coming. Consider expanding this to be more direct about what it means to do this in "forward mode" and "inverse model" It may be more clear to talk about this as "running a forward model" and "solving an inverse problem"? Be consistent with "heat model" or "thermal model" - used interchangeably in title, section headings, and text - but since they mean the same thing just pick one Changed for "solve an inverse problem" and "run the model forward"

Title changed to "heat modelling"

The beginnig of the paragraph is now:

L. 112 : "We present a 1D heat model forced by reconstructed climatic conditions, and run it in two ways. First, we solve an inverse problem with this model to infer the value of the GF in the Dome C region,using radar echoes as observational constraints. The pattern of wet and dry areas allows to estimate a critical ice thickness that corresponds to a threshold between frozen and thawing ice. For a given GF and vertical advection, this thickness is unique, so that we may in turn infer a GF distribution from the pattern

of basal echoes."

Line 96: I think you mean "relationships" instead of "relations"

Correction made, L.124

Line 104: "Characteristic" is more often used than "typical"

Correction made, L. 132

Line 117: "flown" should be "flowed", or better yet "been transported by ice flow" Correction made for "flowed", L. 251

Line 121: "permanently" seems too strong

As "continuous" is characterizing the process, we took out "permanently".

L. 256 : "A continuous hydrological network upstream is only fed by local melting,..."

Section 2.1.3 doesn't add much - why is this a section and why does this get mentioned in this order? It seems like you want to introduce the model equations first, then talk about caveats!

The new version of the "Model" section is reordered in this way :

2.1 Geometry and coordinate system 2.2 Heat equation 2.2.1 Ice thermal properties 2.2.2 Basal boundary conditions 2.2.3 Boundary condition at the surface 2.3 Velocity model 2.4 Proceeding assumptions 2.4.1 Correspondance between wet and thawing areas 2.4.2 GF spatial variability

The last section 2.4.2 is needed as it is one simple but important assumption of the model.

Line 138: typo should be "which" Again, "emulates" seems like the wrong word. Here, maybe "simulates" or "approxi- mates"?

Correction made L. 141. As we answered in the "main concerns", emulates seems to

be the right word.

Line 145: Is D dimensionless? Is K a function of reduced depth? (don't think so)

"Dimensionless" added, L. 154. New ordering of the paper provides information on K just after this section, L. 160. K depends on T, so indirectly on reduced depth.

Equation 2: These might be equivalent or just defined differently, but this equation is not the same as in Parrenin et al. (2007; equation 3) - check to make sure no typos between $\zeta$ and (1-$\zeta$)

No typo here, Parrenin et al 2007 is using reduced height, and we use reduced depths

Section 2.6 header - suggest stating as "Basal boundary conditions"

Correction made L. 165 §2.2.2

Line 187: Why is this an unusual choice?

L187 :"This is an unusual choice for such an important parameter, but we argue that Eq.(9) is consistent with the temperature profile of the EPICA Dome C ice core"

This formulation of the melting point temperature has not been published in a reviewed article yet, only in Ritz's PhD manuscript. It is needed since it seems to better match the measurements at Dome C. However, this choice is discussed later and has no crucial impact on the results (L.396.), now we just say :

L. 171 : "This expression is compatible with the temperature profile at the EDC bore-hole"

Line 202: paleotemperatures aren't "known", they are still estimates

Changed for "estimated" L. 186

Line 209: Why is this represented as 1/6.04K - if report this way make sure to have typed as actual fraction since as it is now with inline slash it could be confusing

We keep 1/6.04 for consistency with literature (Lorius and Merlivat, 1975)

L193 : "In addition to the nominal value 1/6.04 K, we will perform sensitivity studies..."

Section 2.8 - Do you concentrate grid cells near the base, and is vertical spacing good enough? Also, are 1000 year timesteps good enough? Can you say anything about uncertainties related to your solution grid? The current statement mentions there was a tradeoff between accuracy and speed, how much was compromised?

The mesh is regular. The dependence of the result (basal temperature) is of the order of 0.1 K when changing the vertical spacing, and of the order of 0.01 K when changing the timestep, whereas the computation time linearly increases with the number of elements in the mesh.

L. 276 : "The dependence of the basal temperature on discretization is limited to a few tenths of K."

Everywhere you use "till" should be changed to "until" Correction made

Section 3 heading is a bit confusing since it isn't yet clear what is being measured, and it is definitely not heat flux! I would change that to something more clearly related to your data analysis. In general, I suggest coming up with another term for "measurement spots".

L 235 : "3) Measurement spots"

We changed for a longer but more explicit title L285 : "3) Spots where GF will be inferred"

Suggest in the text to remind readers what p is. Same for any variable that was introduced awhile back in the text

"Shape parameter" added on L. 287

Line 269: Is there a short justification for the heat flux range of 40-70 mW/mЁĘ2?

Empirically, the GF values that were found in this range (in fact 45-65 + safety margin). L. 319: "All the GF values were in the range 40-70 mW m -2"

Line 272: Suggest that "inferred" is better term than "derived" Correction made

Line 286: "do not mismatch" - do not match? Or, mismatch?

We confirm that "do not mismatch" is what we intended to say. Since the confidence we have in these points is low, we could expect a mismatch between the surrounding GF field and the value at these points, but it is not the case. We reworded as folllows :

L335 : "The inferred values of the GF nevertheless matched the N-S gradient"

Line 299: I would put your estimated relationship between heat flux and ice thickness in words and remind the reader that this is for the explored range of heat flux values, or does it hold for any heat flux (any melt rate)? Then, the relationship between heat flux and melt rate must correspond to a specific ice thickness, right? I lost the details around equations 15-16 and suggest adding more text to explain what this means and how it was derived.

First we explain the goal of the emulator we need to build :

L. 347 : "As the computation for a given set of parameter lasts several minutes, computing the basal melt rate for each point of our domain would be far too expensive. Here, the result of the whole forward model is mimicked by an emulator that depends on the input parameters H, $\Phi_g$ g and p."

L. 355 : "Over the positive-melt-value domain, we used a least-square minimization method to compute the following relation:"

-Eq. 15-16 : This paragraph is taken off, since it was unnecessarily complicated with respect to what we want to show. Now we simply present the past basal melt rate calculated with the central values of GF and p, which is the main message of the paper.

Line 337: what do you mean by "at the opposite" here?

L 337 : "However, the average basal melt rate is changed at the opposite by 0.1 mm a -1"

When the GF is affected positively by a parameter change through the inverse problem, the melt rate is affected negatively during the forward problem, and vice-versa.

L380 : "However, the average basal melt rate is changed by 0.1 mm a -1 in the reverse direction"

Line 351: Again, not sure how to take this point about an "unusual choice" and why it was chosen that way to begin with

L351 : "Given that the expression of the pressure melting point is an unusual choice in glaciology (Eq. 9), the inverse method was reiterated with a more common value (B = 0.098 K.Pa -1 ) as a test"

As explained in section 2, we choose this dependence of temperature on pressure because it better matches with observation, instead of a widespread expression (for example in Cuffey and Paterson 2010).

L. 399 : "Given that the expression of the pressure melting point is an unusual choice in glaciology (Eq. 9), the inverse method was reiterated with a more common value corresponding to saturated air in the ice (T m = 273.16-0.098 P , Cuffey and Paterson, 2010)"

Line 354: Why is "order of magnitude" good enough?

L. 354 : "The order of magnitude of the results remains the same whatever the expression for the pressure melting point"

"Order of magnitude" was awkward, the sentence is reworded : L. 401 : "The results in terms of basal melting are not significantly affected by the expression of the pressure melting point"

Line 357: The leading sentence doesn't seem to relate to the second sentence and I had to read back to see if I understood what was coming and has already been done. Suggest better lead-in to this section.

A new introduction to this section is proposed :

L.405 :"One way to assess the performance of our model is to compare observed basal state (wet or dry, Zirizzotti et al. (2012)) with the model simulation for present time. For $p = 2$, we compute the $(\Phi_g$ , H c ) relation, corresponding to the present basal state, by sampling H c between 2 700 m and 3 300 m. The two parameters are linked by the following empirical relation:"

Line 365: "build" should be "built", and really do you mean "calculated"? Correction made L. 414

Line 368: What small-scale structures are these E, G, H, I, L locations referring to?

A sentence now introduces specifically the coordinates we refer to (introduction section) :

L. 121 : "For the sake of convenience, the domain is referenced using pairs of letter and numeral, corresponding to the grid of the Italian survey (Fig. 1). In particular, two promising old-ice candidates are located at C6 and H1."

Line 369: Should be "non-melting"

L 369 : "no-melting areas"

L423 : "non-melting areas"

Line 371: Need to rephrase part about ". . . is however often respected . . ." "undoubtly" should be "undoubtedly"

Line 373: What do you mean by "local gap"?

This paragraph is now mainly restated :

L. 415 : "Superimposed with the observation data, the model output shows that large-scale patterns of wet-dry areas are respected, especially on steep bed slopes (candidate B, C, D, and to a lesser extent candidate A). On these bed reliefs, certain points however show a discrepancy between model and observation, but the gap to the critical thickness is often close to 0 m (D3, D5, D8, M3), meaning thata small change in GF forcing, or a better description of the ice thickness, would better assign theseparticular points. The 1 km-resolution of the Bedmap 2 bedrock dataset (Fretwell and coauthors, 2013) smoothed along-track subkilometric features detected by our RES survey.

The steeper the bed, the sharper the limit between melting and non-melting areas. In the central, flatter, part of the domain, the position of this limit is more blurred. As we could not assess the GF with our method there, it was interpolated. Despite this lack of constraints, several small-scale features are well mimicked (dry areas at I9, G-H8, wet areas at G9 and L7). Other regions are not assigned in compliance with observations (G6-7-8, H-I8), meaning that the GF is overestimated, probably up to 3mW m -2 , which is consistent with the uncertainties given by our method (inversion and interpolation)."

Line 375: Should be "dependent" Correction made L. 432

Section 6.1, maybe "Method validation"? There is no lead in, which may be fine but as is now it isn't clear how come you need to have overarching section 6.1

The structure of the discussion section has now changed, and we changed for "Model assessment"

6.1 Consistency with published data and measurements 6.2 Model assessment 6.2.1 Method validity 6.2.2 Sensitivity to parameters 6.2.3 Spatial variation of the GF field 6.3 Lessons drawn for the oldest-ice research 6.3.1 Interpretation of the wet/dry pattern at the ice base 6.3.2 Old-ice targets

Line 399: What do you mean "the surface slope is the source of motion"? Talking about clues isn't very precise - is there a better word?

[Figure]

At first order there is motion because of the pressure gradients within the ice, and that depends on surface slope. These explanations are now put in section 2.4.1.

L. 260 : "For a given ice flux, the surface slope gives some indication on the relative importance of internal deformation and basal sliding in the ice motion. Local steeper surface may be associated with more basal drag and ice deformation, whereas local flatter surface may be associated with more sliding."

Paragraph around lines 415-420: Not sure I understood the point of this paragraph, if this is just too hard to constrain state that directly. If it needs to be considered somehow and will significantly affect uncertainty estimate state that directly too. I can't tell if this is something that really could matters.

This paragraph (now L486) correspond to additional tests we made. We wanted to make sure of the influence of accumulation, since there is a NS gradient of both accumulation and GF. As we made the sensitivity test for a, it is worth saying a word about it, even if the sensitivity is low.

Line 422: Not sure what you mean with "litmus"

"Litmus test" means a "test of truth" (changed for "reliable comparison" L. 443).

Section 6.1.4 - by "structure" do you mean "spatial variation"

Correction made for "6.2.3 Spatial variations of the GF field" L. 492

FIGURES AND TABLES Would be worth defining input parameters in Table 2. I don't quite understand what "total on m" represents.

The new caption is the following :

"Sensitivity of the GF (assessed from inverse runs, [mW m -2 ]) and basal melt rate (calculated with forward runs, [mm a -1]) to input parameters $\alpha$, $\beta$ and a. As the final value of m depends on both the inverse run to determine $\Phi_g$, and the forward run to compute the melt rate, the last column accounts for the sensitivity on the whole

procedure (inverse+forward)."

As the final value of m depends on both the inverse run to determine the GF, and the forward run to compute the melt rate, the last column accounts for the sensitivity with the whole method (inverse+forward), which is less than the sensitivity with the forward mode only.

Figure 2: I might have missed it but check that critical ice thickness is defined in the text to this point, or add to caption

Caption completed :

"We define the critical thickness as the minimum thickness that allows basal melting at present."

Figure 4: Not sure it was discussed in the text how the 10 spots were chosen?

The spots are chosen where a correlation could be found between reflectivity and ice thickness. Sentence modified in section 3 :

L. 293 : "Ten corresponding spots are selected (black rectangles in Fig. 1), where reflectivity and ice thickness are somehow correlated, and that are hereafter denoted by the indexes of their central point (Fig. 4)."

Figure 6: A lot of overlapping lines. How many discrete values of m are represented?

Now only 2 series of m-contours are represented, for p=1 and p=10.

---

## Author Comment (AC3) · 19 Jun 2017

Thank you for this short comment.

Our results are indeed quite close to the results of Siegert and Dowdeswell (1996), and a reference to this paper is added in the discussion section.

---

## Author Response (AR2)

SUGGESTIONS:

P9-10: Is it better to say "We used an inverse approach to retrieve GF from radar-inferred distribution of wet and dry beds"? The current writing sounds like that the authors made a complicated analysis of the radar data in their own work to tie the magnitude of the bed reflectivity and GF.
We agree with this more specific formulation, and changed for the proposed sentence.

L12: local distribution "of radar-inferred basal melting"
Changed

L15-16: It is said "two main subregions" at L15 but then "a third one" at L16.
The sentence is reworded this way :
"Three main subregions appear to be free of basal melting, two because of a thin overlying ice, and one north of Dome C, because of a low GF."

L49: "parameter's"
Changed

L68: typo? "are rare in Antarctica…" Delete "East"
Changed

L70: Add a reference supporting a typical length scale of 10 km.
We now explicit this comes from the work of Carson et al 2014.

L70-75: Melting spots can be found as spots with anomalously high bed return power, only if the study area includes both melting and freezing regions. If the entire study area has melting bed, then the radar data are not necessarily capable to show the melting spots.

The new sentence is specified by  a new beginning :
Over a study area covered both by frozen and melting basal ice, radio echo sounding (RES) measurements may help infer the basal conditions at regional scale, since the presence of water at the ice-bed interface is responsible for a remarkable increase in the amplitude of the reflected echoes.

L110: add "today" after "melting occurs"
Changed

L145: remove the first "and"
We rather think a comma was missing :
"(in particular bedrock and surface heights, and ice thickness)"

L428: what is the basis to argue "probably up to 3 mW/m2" here?
The sentence is split and specification is added :
"As the difference in critical ice thickness does not generally exceed 150 m, and as 1 mW m -2 has the same thermal effect as 60 m in ice thickness, the GF is probably overestimated up to 2.5 mW m -2, which is consistent with the uncertainties produced by our method (inversion and interpolation)."

L436-L437: Do you mean "Hence, considering the duration of deglaciation and strong dependence

on the ice thickness, the thermal state of the basal ice may ….”?
The sentence is reworded this way :
“Hence, considering the duration of deglaciation, the thermal state of the basal ice may correspond to very different climatic periods depending on the ice thickness”

L445: “positive GF anomaly”
Changed

L469-470: Do you mean “highest (with thinnest ice) DRY points and the lowest (with thickest ice) WET points”?
No, we mean the highest point (with thinnest ice) for which the conditions are *still wet*, and the lowest point (with thickest ice) for which the condition are *still dry*. We here refer to the way the critical thickness is evaluated, so we just add to refer to §3.

“between the highest wet points and the lowest dry points at a given spot (see §3).”

L532: will now be performed?? Do you mean “is being performed” or “will be performed soon”?
Changed for “Is now being performed”

L577: “state”
Changed

Table 1 caption: GHF -> GF
Changed

Table 1: make the column header consistent; Hc should be Hc +/- sigma.
Changed

Fig. 1 caption: change the second dry to “(red)” in the first line.
Changed

Fig. 2: GHF to GF.
Changed

Fig 3: Add “shape parameter” in front of the italic p.
Changed

Fig. 5: change to “Geothermal flux”. Also change the legend in the figure accordingly.
Changed

Fig. 6: Change the x axis. Add italic “m” after Basal melt rate”. Again, geothermal flux. Add “shape parameter” in front of the italic p.
Changed

Fig. 7: “Averaged past” remains not clear enough.
Specified this way : “Past basal melt rate, averaged over 400 ka.”

Fig. 8: Geothermal flux.
Changed

L12, 35, 46, 52, …, 548 (and at many other places): Geothermal flux -> GF
Occurrences checked and changed

 Non-public comments to the Author:
 It is completely up to the authors, but I feel very happy if the authors acknowledge reviewers'
 work explicitly in the paper. Sorry for slow process to handle this paper, while I was absent from
 the office and with very narrow internet at a summer destination.

Be sure that we had no bad intent ! We just waited the end of the editing process to thank every
person and organization that allowed this work to be lead. Now complete acknowledgement is
added.